

# Integrability and complexity in quantum spin chains

Ben Craps[1★], Marine De Clerck[2†], Oleg Evnin[3,1‡] and Philip Hacker[1∘]

**1** Theoretische Natuurkunde, Vrije Universiteit Brussel (VUB) and
The International Solvay Institutes, Pleinlaan 2, B-1050 Brussels, Belgium
**2** Department of Applied Mathematics and Theoretical Physics, University of Cambridge,
Wilberforce Road, Cambridge CB3 0WA, United Kingdom
**3** High Energy Physics Research Unit, Faculty of Science, Chulalongkorn University,
Thanon Phayathai, Bangkok 10330, Thailand

★ Ben.Craps@vub.be , † md989@cam.ac.uk ,
‡ oleg.evnin@gmail.com , ∘ philiphacker1@gmail.com

## Abstract

There is a widespread perception that dynamical evolution of integrable systems should be simpler in a quantifiable sense than the evolution of generic systems, though demonstrating this relation between integrability and reduced complexity in practice has remained elusive. We provide a connection of this sort by constructing a specific matrix in terms of the eigenvectors of a given quantum Hamiltonian. The null eigenvalues of this matrix are in one-to-one correspondence with conserved quantities that have simple locality properties (a hallmark of integrability). The typical magnitude of the eigenvalues, on the other hand, controls an explicit bound on Nielsen's complexity of the quantum evolution operator, defined in terms of the same locality specifications. We demonstrate how this connection works in a few concrete examples of quantum spin chains that possess diverse arrays of highly structured conservation laws mandated by integrability.



# 1 Introduction and statement of results

One often thinks that solving problems makes them easier. Among solvable dynamical problems in theoretical physics, those described by integrable systems appear prominently. Is there a quantifiable sense in which the dynamics of integrable systems is 'easier' than the dynamics of generic systems?

Recent years have seen a few different complexity measures applied to quantum evolution operators. Our focus here will be on Nielsen's complexity [1–3] defined for unitary operators in terms of geodesics on the group of unitaries endowed with an appropriate physically motivated metric. This notion of complexity has been applied to quantum evolution operators in [4–6]. There is a related branch of research that appeals to Krylov complexity [7,8] in an attempt to quantify how many different operators/states one needs to approximate well the time evolution starting from a given operator/state.[1] While in all of these approaches, some difference has been observed between complexities of integrable and chaotic evolutions [5, 6, 17], no direct relation between the integrability structure (the presence of a large number of analytic conservation laws) and complexity reduction has been presented (see also [18]). Our aim in this article is to close this gap.

Nielsen's complexity enjoys a simple geometric definition and a close relation to the notions of computational complexity considered in the field of quantum information. It is, however, notoriously difficult to compute in practice, since there is no effective way to find globally optimal geodesics on the manifold of unitaries associated with high-dimensional Hilbert spaces.

---

[1]The study of Krylov complexity was recently extended to periodically driven quantum systems in [9]. The complexity of quantum states, with special attention paid to the simple case of harmonic oscillators, has been investigated in [10,11] and its connections to other measures of chaos are examined in e.g. [12–14]. We mention additionally the interesting considerations in [15] that apply quantities derived from quantum resource theory to the evolution of integrable and chaotic systems. A relation between Krylov and Nielsen's complexity has been described in [16].

To bypass this difficulty, we asked in [6] whether simpler quantities can be constructed that, on the one hand, provide a bound on the actual Nielsen's complexity and, on the other hand, can be effectively computed. A key further concern was whether such a bound can be powerful enough in itself to distinguish integrable and chaotic evolution. The answer to both questions was in the affirmative. First, an upper bound on Nielsen's complexity could be constructed, where finding the optimal geodesic on the manifold of unitaries was traded for minimizing a multivariate quadratic polynomial over an integer hypercubic lattice. (The latter problem is still known to be very hard, but a range of highly effective approximate minimization algorithms has been developed, mostly in relation to problems of 'lattice cryptography.') Second, the bound constructed in this way showed sensitivity to integrability vs. chaos in a range of trials based on explicit quantum Hamiltonians.

It is worthwhile to make a short digression here and compare the findings of [6] to the results of a similar investigation undertaken in [17] for the case of Krylov complexity. The reasoning behind Krylov complexity is very different from Nielsen's complexity. It is not directly rooted in a relation to quantum information and quantum computation, but rather adapted from the start to quantum evolution, quantifying the speed with which the operator evolution explores different directions in the space of operators. An advantage of Krylov complexity is that it can be evaluated directly and with only modest computational cost. In [17], the behavior of Krylov complexity was investigated for a range of quantum systems, and it was demonstrated that integrable evolution indeed generates some reduction in Krylov complexity. The typical scale of this reduction is of order one, consistent with the findings in [6] for the upper bound on Nielsen's complexity. While much stronger complexity reduction (from exponential to polynomial in the entropy) has occasionally been anticipated for integrable systems [5], there is no rigorous backing for such views. For Nielsen's complexity, hopes remain that the actual quantity, rather than the upper bound considered in [6] will show a stronger reduction for integrable systems. An illuminating observation of [17] is that the reduction of Krylov complexity in integrable systems can be linked to the larger variance in this case of the Lanczos coefficients inherent in the construction of Krylov complexity. This larger variance appears to diminish the complexity growth via mechanisms analogous to Anderson localization.

The observations regarding the reduction of complexity for integrable systems (for Krylov complexity in [17] and for Nielsen's complexity in [6]) are reassuring, but they have been phrased as empirical observations for concrete systems. For Krylov complexity, the relation to Anderson localization has provided a nice qualitative picture, but establishing a precise relation between the large variance of Lanczos coefficients that underlies this property on the one hand and integrability on the other hand remains an outstanding problem. Our purpose in this paper is to develop an analytic connection between integrability and complexity reduction in the context of the upper bound on Nielsen's complexity introduced in [6].

As we have already mentioned, evaluating the upper bound on Nielsen's complexity developed in [6] amounts to minimizing a multivariate quadratic polynomial over a hypercubic lattice. The quadratic polynomial is defined through a matrix, called the $Q$-matrix in [6], which is in turn constructed from the energy eigenvectors of the quantum system under study. It is this matrix that provides a bridge between the notions of integrability and complexity. More precisely, the typical magnitude of the eigenvalues of $Q$ controls the magnitude of the polynomial minimized over the lattice, and thereby controls the estimate for complexity. At the same time, the null eigenvalues of $Q$ correspond to conservation laws (more precisely, the conservation laws that lie within the subspace of 'easy' operators inherent in the definition of Nielsen's complexity). Since integrable systems are precisely defined by large numbers of conservation laws, this immediately introduces a large number of null eigenvalues of $Q$. This, first, directly lowers the average magnitude of the eigenvalues of $Q$, immediately affecting the complexity. Furthermore, for a large matrix, one can naturally expect that the spectrum

displays some level of continuity, and raising the number of null eigenvalues comes together with raising the number of small non-zero eigenvalues, lowering the complexity even further. Thus, the *Q*-matrix provides a quantitative connection between integrability and complexity that has long been elusive.

To explore these ideas, we will turn to quantum spin chains [19]. This setting is advantageous in the sense that various spin chains possess diverse integrable structures with towers of conservation laws displaying a range of behaviors. The definition of Nielsen's complexity depends on a choice of 'easy' directions in the space of operators, and different choices lead to different complexity estimates. For example, one can designate as 'easy' those operators that are built as products of no more than a given number of single-site spin operators, or introduce further restriction like the spatial extent of lattice sites contributing to the given operator. The number of conservation laws that fit each of these classes is different, and one can track how it correlates with the complexity estimates. Analyzing these connections will be our main practical goal.

As a counterpoint to our studies of complexity in the presence of integrability, it is interesting to also develop a picture of how the *Q*-matrix (and therefore the complexity bound of [6]) behaves for generic systems, where the set of energy eigenvectors will be assumed to be a random basis (while the 'easy' directions in the set of operators are prescribed). In this situation, tools from random matrix theory (RMT) can be used to deduce complexity estimates that will be further compared to the results of our numerical experiments.

## 1.1 Statement of results

The bound on Nielsen's complexity that shall be our main tool for estimating the complexity of the evolution operator of dynamical systems takes the form of a discrete minimization at every time step $t$:

$$\mathcal{C}_{\text{bound}}(t) = \min_{\mathbf{k} \in \mathbb{Z}^D} \left\{ \sum_{mn} (E_n t - 2\pi k_n) \big[ \delta_{nm} + (\mu - 1) Q_{nm} \big] (E_m t - 2\pi k_m) \right\}^{1/2}, \qquad (1)$$

with

$$Q_{nm} \equiv \sum_{\dot{\alpha}} \langle n|T_{\dot{\alpha}}|n\rangle \langle m|T_{\dot{\alpha}}|m\rangle = \delta_{nm} - \sum_{\alpha} \langle n|T_{\alpha}|n\rangle \langle m|T_{\alpha}|m\rangle . \qquad (2)$$

Here $E_n$ and $|n\rangle$ are the eigenvalues and eigenvectors of the Hamiltonian, and $\{T_\alpha\}$ ($\{T_{\dot{\alpha}}\}$) a basis of generators for the 'easy' ('hard') directions on the manifold of unitaries. The parameter $\mu$ defines the increased cost in the length functional associated to the hard directions. The bound (1) was obtained in [6] by restricting the original minimization problem relevant to Nielsen's complexity to geodesics of the natural metric on $U(D)$ without penalties (at $\mu = 1$) and was observed to distinguish chaotic from integrable dynamics.

The main results of the present work can be summarized in three points. First, we propose an algorithmic procedure to extend the bound (1) in an effective way to degenerate energy spectra. Second, we derive an estimate for the late-time saturation value of (1) for generic chaotic models relying on random matrix theory. We find that it is determined by the mean of the eigenvalue distribution of the corresponding *Q*-matrix (2) and, moreover, can be qualitatively predicted by the ratio of the number of easy generators $T_\alpha$ to the total number of generators. Third, we propose a quantitative connection between integrability and complexity reduction. We identify and provide substantial evidence for a mechanism by which integrability lowers complexity. In the language of the bound (1), this mechanism finds its origin in the emergence of non-costly directions on the space of unitaries realized by zero (and small) eigenvalues of the *Q*-matrix (2), which, as we mentioned above, are a direct consequence of the locality properties of the towers of conservation laws in integrable systems. The proposed

connection is verified in great detail in quantum spin chains, by investigating and validating the insensitivity of the complexity measure (1) to systematic modifications of the set of 'easy' directions that preserve the locality properties of the conservation laws (and hence, the number of zero $Q$-eigenvalues), as well as its sensitivity to those modifications that alter these locality properties.

The paper is structured as follows: The methods and techniques developed in [6] to obtain the practical upper bound (1) on Nielsen's complexity for finite-dimensional quantum systems will be reviewed in section 2.1 and generalized to systems with a degenerate energy spectrum in section 2.2. In section 3, we apply RMT as a tool to extract general properties of the distribution of $Q$-eigenvalues as well as the late-time complexity plateau height for generic chaotic Hamiltonians in the limit of large matrix dimension. Finally, section 4 presents our analysis of Nielsen's complexity in integrable and chaotic spin chain models and discusses correlations between the number of local conservation laws and complexity estimates.

## 2 A computable upper bound on Nielsen's complexity

In section 2.1, we review the variational Ansatz formulated in [6] that yields an upper bound on Nielsen's complexity for the time-evolution operator of a quantum system. Originally, this Ansatz was developed under the assumption that the spectrum of the Hamiltonian is nondegenerate. In the presence of degeneracies, however, an ambiguity appears because a choice of energy eigenbasis is needed as an input in the procedure. Since the focus of section 4 shall be on spin chains, which often display degenerate energies due to global symmetries, we develop a strategy to resolve this ambiguity in section 2.2.

### 2.1 Bounding Nielsen's complexity: A review

#### 2.1.1 Definition of Nielsen's complexity for Hamiltonian evolution

Consider a quantum system described by a Hamiltonian $H$ acting on a $D$-dimensional Hilbert space $\mathcal{H}$. We are interested in evaluating the complexity of the associated evolution operator $e^{-itH}$ as a function of time. Following the original definition by Nielsen [1] and the recent applications to quantum evolution in [4, 5], one characterization of the evolution operator complexity at a fixed instant $t$ is given by the length of the shortest path[2] $U : [0, t_f] \to U(D)$ in the group of unitaries $U(D)$ starting at the identity $\mathbf{I}$ and ending at $e^{-itH}$

$$U(0) = \mathbf{I}, \qquad U(t_f) = e^{-itH}. \tag{3}$$

An essential part of the definition of Nielsen's complexity involves a choice of the metric on $U(D)$. From a quantum computation perspective, it is natural to adopt a distance measure that favors some directions over others since some operations on the system may be harder to implement than others. This intuition is borrowed from quantum circuit complexity where one aims to understand how to most efficiently compose a unitary acting on a collection of qubits with a restricted set of available simple operators (called quantum gates) that act on at most a few qubits at a time.

For a generic quantum system, this heuristic picture can be implemented by starting with a basis of normalized generators $T_i$ for the tangent space at every point on $U(D)$,

$$\mathrm{Tr}[T_i T_j] = \delta_{ij}, \tag{4}$$

---

[2]Without loss of generality, in the following we shall fix the endpoint of the parameterization of the curve $U(t')$ by setting $t_f = t$. This can always be achieved by simultaneously rescaling $V \to Va$ and $t' \to t'/a$ by a constant value $a$.

and splitting the generators into two groups. The generators associated with the directions in which curves can run at a low cost shall be denoted by $T_\alpha$ and the costly directions by $T_{\dot\alpha}$. This partitioning is usually physically motivated and chosen according to the locality properties of the generators. The locality degree of an operator can often be specified with a single number $k$ or a small set of numbers $\{k_a\}$. The easy directions $T_\alpha$ are then chosen in such a way that their locality degree does not exceed a certain maximal locality degree $k_{\max}$. In qubit systems, for instance, three or higher qubit gates are conventionally considered hard [1,2]. More generally, the locality of the Hamiltonian, which is typically made out of few-body operators, represents a natural locality threshold. This convention was adopted in e.g. [4,5]. Alternatively, the locality threshold can be treated as a free parameter which allows one to investigate how the complexity varies with $k_{\max}$ [6]. On physical grounds, $k_{\max}$ should nevertheless be thought of as much smaller than the number of degrees of freedom in the system (such as e.g. the total number of spins). For our purposes, we shall always assume that the Hamiltonian can be expanded as a function of the easy generators $T_\alpha$ alone.

We proceed to discuss in more detail the construction of a metric that predominantly drives the geodesics along the easy directions $T_\alpha$, following [1,33,34]. This anisotropic feature in the geometry of $U(D)$ can be realized by a distance measure between two nearby unitaries $U$ and $U + dU$ of the type

$$ds^2 = \sum_{kl} \mathrm{Tr}[i\, dU U^\dagger T_k]\, g_{kl}\, \mathrm{Tr}[i\, dU U^\dagger T_l], \tag{5}$$

where the matrix $g$ is designed to penalize the 'hard' directions by increasing the magnitude of the corresponding matrix entries. In the following, we shall restrict ourselves[3] to situations where the costly directions share a common penalty factor $\mu$

$$g = \begin{pmatrix} \delta_{\alpha\beta} & 0 \\ 0 & \mu\, \delta_{\dot\alpha\dot\beta} \end{pmatrix}. \tag{6}$$

The expression (5) is often termed right-invariant because it is invariant under right multiplication (but not under left multiplication) of $U$ with any unitary. In the absence of a penalty for $T_{\dot\alpha}$, i.e. when $\mu = 1$, the metric (5) reduces to the standard bi-invariant metric on $U(D)$

$$ds^2_{\text{bi-inv}} = \mathrm{Tr}[dU^\dagger dU]. \tag{7}$$

The complexity metric (5) appears more intuitive when introducing the Hermitian velocity operator $V(t')$ along a curve $U(t')$ on $U(D)$

$$V = i\frac{dU}{dt'}U^\dagger, \tag{8}$$

whose coefficients in terms of the easy and hard generators (4) are given by

$$V^\alpha \equiv \mathrm{Tr}[V T_\alpha], \quad V^{\dot\alpha} \equiv \mathrm{Tr}[V T_{\dot\alpha}]. \tag{9}$$

Evaluated on a curve with velocity (8), the metric (5) takes on a simple form

$$ds^2 = dt'^2 \Big( \sum_\alpha (V^\alpha(t'))^2 + \mu \sum_{\dot\alpha} (V^{\dot\alpha}(t'))^2 \Big), \tag{10}$$

in which the contributions associated with the hard directions are weighted with the penalty factor $\mu$. Equipped with this 'complexity metric', Nielsen's complexity of the evolution operator

---

[3]See e.g. [33,35] for works discussing more convoluted penalty choices.

corresponds to the length[4] of the path with boundary conditions (3) and velocity (8) that optimizes the length defined with (10):

$$\mathcal{C}(t) = \min_{V} \int_{0}^{t} dt' \Big[ \sum_{\alpha} \big(V^{\alpha}(t')\big)^2 + \mu \sum_{\dot{\alpha}} \big(V^{\dot{\alpha}}(t')\big)^2 \Big]^{1/2}. \tag{11}$$

A geometric broad approach to characterizing the complexity of unitary operators has several advantages over quantum circuit complexity. First, this geometric perspective allows for new tools from the well-studied area of Riemannian geometry to study the complexity of quantum operations. Second, the length-minimizing paths in the metric (10) solve a second order differential equation and are therefore uniquely defined after specifying an initial position and velocity. The local condition imposed by the geodesic equation can be contrasted with the absence of any relation between the successive unitaries in the optimal discrete quantum circuit implementing a desired unitary operator.

The right-invariant metric (10) was initially proposed, together with other Finsler metrics on $U(D)$, to provide a lower bound on quantum circuit complexity in qubit systems [1]. A subsequent work [2] showed that Nielsen's complexity was in fact polynomially equivalent to approximate quantum circuit complexity, where a unitary is implemented by a chain of (universal) local gates to a very good approximation, provided the cost factor is taken large enough. The precise scaling $\mu(D)$ required for the equivalence between the two notions to be valid is however dependent on the complexity itself [2,5]. Relying on the assumption that the complexity of the unitary operator increases polynomially with the size of the Hilbert space, the cost factor $\mu$ has been conventionally chosen to scale linearly with $D$ in the past [4,5].

Unlike the original circuit complexity, Nielsen's complexity is very well-adapted to discussing continuous evolution processes. For that reason, as far as the complexity of quantum evolution is concerned, the relation between Nielsen's complexity and circuit complexity is less relevant and should merely be understood as motivational. From a physical point of view, $\mu$ should only be constrained to be large enough so as to lead the geodesics through the valleys created by the local directions. As we shall discuss in detail, in the context of our variational Ansatz, the exact scaling of $\mu$ with $D$ will be irrelevant. As long as $\mu$ is large, the plateau value of the upper bound on the complexity of generic chaotic models will scale as $\mathcal{C} \sim \sqrt{\mu D}$. For definiteness, we shall set $\mu = D$ in our subsequent numerics, as in [6].

Varying the cost factor $\mu$ modifies the geometry and, as a consequence, the geodesic solutions whose lengths enter the complexity (11). In particular, in the limit $\mu \to \infty$, the geodesics are only allowed to run in purely local directions. This limit in fact most closely resembles the framework of exact quantum circuit complexity. The geometry it defines is known as sub-Riemannian geometry [20–22]. It is known that the length-minimizing curves converge in the $\mu \to \infty$ limit to piecewise-smooth curves that run exclusively in the 'easy' directions.

### 2.1.2 Solving bi-invariant complexity

The solutions to the geodesic equation for the metric (5) are hard to find for a generic value of $\mu$. However, when $\mu = 1$, every direction is treated equally,

$$ds^2_{\text{bi-inv}} = \text{Tr}[dU^{\dagger}dU] = dt'^2 \left( \sum_{i} \big(V^{i}(t')\big)^2 \right), \tag{12}$$

---

[4]Nielsen's original definition for the complexity differs from (11) by an overall factor of $1/\sqrt{D}$. One can straightforwardly translate our results to Nielsen's convention by simply rescaling the vertical axis of all complexity plots and complexity plateau heights by this factor.

and the associated complexity takes the simple form

$$\mathcal{C}_{\text{bi-inv}}(t) = \min \int_0^t dt' \Big[ \sum_i \big(V^i(t')\big)^2 \Big]^{1/2}, \tag{13}$$

where the sums run over all the directions $T_i$ on $U(D)$. While the notion of bi-invariant complexity is of little physical interest by itself, reviewing this elementary case first provides a good pedagogical preview of the structures relevant for our later treatment of Nielsen's complexity at $\mu \neq 1$.

When all the generators are on the same footing, the geodesic equation is very simple and given by $dV^i(t)/dt = 0$. The solutions are curves of constant velocity $V$,

$$U(t') = e^{-iVt'}. \tag{14}$$

The geodesics of interest to Nielsen's complexity are required to connect the identity operator to the evolution operator $e^{-itH}$. Hitting $e^{-itH}$ at $t' = t$ imposes the condition

$$e^{-iVt} = e^{-iHt}. \tag{15}$$

For nondegenerate energy spectra, at any instant of time $t$, only a discrete, though infinite, family of these bi-invariant geodesics play a role in extremizing (13). In fact, it straightforwardly follows that the family of candidate geodesics for minimizing (13) is parameterized by a $D$-dimensional vector of integers $\mathbf{k}$

$$U_{\mathbf{k}}(t') = \exp(-iV_{\mathbf{k}} t'), \tag{16}$$

with

$$V_{\mathbf{k}} = \sum_n \Big( E_n - \frac{2\pi}{t} k_n \Big) |n\rangle\langle n|, \tag{17}$$

where $E_n$ and $|n\rangle$ are respectively the Hamiltonian's eigenvalues and eigenvectors. In the presence of degenerate energy levels, the constraint (15) is solved by a larger set of velocities, since there exists a continuum of different choices for the energy eigenbasis. As a result, the velocities associated to curves which satisfy the correct boundary conditions are parameterized by continuous angles, in addition to the discrete integers $k_n$. We shall defer a more detailed treatment of the general situation to section 2.2 and assume for now that the energy spectrum is not degenerate.

The length of the 'toroidal' curve (16) is controlled by the trace-norm of its constant velocity $V_{\mathbf{k}}$, such that the bi-invariant complexity can be written as

$$\mathcal{C}_{\text{bi-inv}}(t) = \min_{\mathbf{k} \in \mathbb{Z}^D} \Big[ \sum_n (t E_n - 2\pi k_n)^2 \Big]^{1/2} \equiv \min_{\mathbf{k} \in \mathbb{Z}^D} \Big[ \sum_n E_n'^2 \Big]^{1/2}. \tag{18}$$

Note that the geodesics (16)-(17) depend on the time $t$ at which the evolution operator $e^{-iHt}$ is evaluated.

The discrete minimization problem (18) is solved at any time $t$ by choosing $k_n$ such that $E_n' \in [-\pi, \pi[$. Geometrically, one can think of the real vector $\mathbf{E}t$ extending as time evolves with a constant slope through a $D$-dimensional hypercubic lattice of spacing equal to $2\pi$. The minimization in (18) is then straightforwardly solved at any instant of time by projecting the vector $\mathbf{E}t$ onto the closest lattice site in $(2\pi\mathbb{Z})^D$. This can be achieved by rounding each component $E_n t/(2\pi)$ to the nearest integer

$$k_n = \lfloor E_n t/2\pi \rceil. \tag{19}$$

Evidently, as long as $t$ is smaller than $\pi/|E_n|$ for any $n$, the nearest lattice point lies at the origin with $k_n = 0$ and the shortest geodesic connecting $e^{-iHt}$ to the identity is the quantum evolution itself. The early-time complexity then increases linearly with time

$$\mathcal{C}_{\text{early}}(t) = t \left( \text{Tr}[H^2] \right)^{1/2} = t \left( \sum_n E_n^2 \right)^{1/2} , \tag{20}$$

with a slope set by the norm of the Hamiltonian. As in [6],[5] in the remainder of this paper we shall adopt the following normalization for the Hamiltonian operator

$$\text{Tr}[H^2] = \sum_n E_n^2 = 1 , \tag{21}$$

which can be attained by rescaling the time parameter. Using this convention, the complexity displays a universal unit slope at early times

$$\mathcal{C}_{\text{early}}(t) = t , \tag{22}$$

for all quantum systems and independent of the dimension of the Hilbert space. We shall discuss below that this early time linear growth remains valid when turning on the cost factor $\mu$. Physically, this property of Nielsen's complexity reflects the simple fact that, at early times, any system is most efficiently simulated by its own evolution.

The initial linear growth is modified at $t = \pi/|E_n^{\max}|$, when the minimizer of (18) becomes the integer vector $\mathbf{k}$ that has a single 1 in the direction corresponding to the largest (absolute) eigenvalue $E_n^{\max}$. In particular, the curve traced by $e^{-iHt}$ only remains length-minimizing for a time interval set by the energy of maximal absolute value. Since the manifold of unitaries is compact, the optimal distance between any two unitary operators, and hence the complexity, is bounded from above. Within our Ansatz, (18) is manifestly bounded from above by $\sqrt{D}\pi$ since every term in the sum can be chosen to lie between $-\pi$ and $\pi$ using the optimal $k_n$ given by (19). However, this rough upper bound typically overestimates the saturation value of the complexity [6]. The reason for this is most easily understood in the geometric picture sketched above (19). At later times, the vector $\mathbf{E}t$ can be expected to be a typical point in $\mathbb{R}^D$. With this assumption, the late-time saturation value of Nielsen's complexity is governed by the typical distance between $\mathbf{E}t$ and the hypercubic lattice $(2\pi\mathbb{Z})^D$. This typical distance can be estimated in the large $D$ limit [6, 23] and yields

$$\mathcal{C}_{\text{bi-inv}}^{\text{sat}}(t) = \sqrt{\frac{D}{3}}\pi . \tag{23}$$

The prediction for the saturation value of (18), as well as the fluctuations about the plateau, were verified in detail in [6]. This bi-invariant version of Nielsen's complexity was previously analyzed in [4, 24] for chaotic models. In [6], the time evolution of the complexity was shown to be mostly similar for integrable and chaotic models, with subtle differences at intermediate times. In particular, the height of the plateau region of the bi-invariant complexity curves is in general not sensitive to the dynamics of the model.

### 2.1.3  Algorithmic approach to bounding Nielsen's complexity at $\mu > 1$

We now compare this story to the more physically interesting case of Nielsen's complexity with a large penalty factor, following [6]. Specifically, we consider the problem of solving (11)

---

[5]In [6], we additionally imposed a tracelessness condition on the Hamiltonian. In the present context, we will be focused on spin chain Hamiltonians which are usually traceless by construction.

with $\mu = D$. Changing the value of the cost factor modifies the geometry and the corresponding geodesics altogether. As mentioned above, for any sizeable number of dimensions, the geodesics of (5) are much harder to compute when $\mu \neq 1$. The toroidal bi-invariant geodesics (14) nonetheless continue to make sense as paths connecting two given points. Therefore, a natural 'variational' approach to constraining Nielsen's complexity at $\mu > 1$ consists in restricting the full minimization in (11) to a minimization over the discrete family of curves (16). Although the bi-invariant geodesics (16) are generically no longer good candidates to minimize exactly the complexity functional (11) at $\mu \neq 1$, their simple shape and the straightforward expression for their length provide a practical way to bound from above the length of the true global geodesic connecting the identity and the evolution operator. This variational approach to the initial minimization problem (11) hence allows for the derivation of an upper bound on Nielsen's complexity in the presence of penalties. There is generically[6] no guarantee that the upper bound is close to Nielsen's complexity,[7] but this simple-minded approach succeeds in differentiating chaotic from integrable dynamics [6] and therefore represents an interesting dynamical probe on its own.

Restricting the minimization in (11) over the curves (16), one finds

$$\mathcal{C}_{\text{bound}}(t) = t \min_{\mathbf{k} \in \mathbb{Z}^D} \Big[ \sum_\alpha \big(V_\mathbf{k}^\alpha\big)^2 + \mu \sum_{\dot\alpha} \big(V_\mathbf{k}^{\dot\alpha}\big)^2 \Big]^{1/2} = t \min_{\mathbf{k} \in \mathbb{Z}^D} \Big[ \sum_i \big(V_\mathbf{k}^i\big)^2 + (\mu-1) \sum_{\dot\alpha} \big(V_\mathbf{k}^{\dot\alpha}\big)^2 \Big]^{1/2}, \tag{24}$$

where the sum over $i$ runs over both the local and nonlocal generators of $U(D)$. Expression (24) has an elegant geometric interpretation, which parallels our discussion of the bi-invariant case. To see this, we start by noting that

$$\delta_{mn} = \text{Tr}\,(|n\rangle\langle n|m\rangle\langle m|) = \sum_i \langle n|T_i|n\rangle\langle m|T_i|m\rangle. \tag{25}$$

The second equality follows from expanding the projectors in terms of the generators $T_i$ and using the normalization (4). Then, combining (25) with the coefficients (9) of the velocities (17), (24) becomes

$$\mathcal{C}_{\text{bound}}(t) = \min_{\mathbf{k} \in \mathbb{Z}^D} \Big\{ \sum_{mn} (E_n t - 2\pi k_n) \big[\delta_{nm} + (\mu-1)Q_{nm}\big] (E_m t - 2\pi k_m) \Big\}^{1/2}, \tag{26}$$

with

$$Q_{nm} \equiv \sum_{\dot\alpha} \langle n|T_{\dot\alpha}|n\rangle\langle m|T_{\dot\alpha}|m\rangle = \delta_{nm} - \sum_\alpha \langle n|T_\alpha|n\rangle\langle m|T_\alpha|m\rangle. \tag{27}$$

We hence obtain a geometric picture similar to the bi-invariant complexity, where (26) takes on the form of the minimal distance between the hypercubic lattice $(2\pi\mathbb{Z})^D$ and a vector which grows linearly with time in a direction set by the vector of energy eigenvalues $\mathbf{E}$. In the presence of a cost factor, however, the distance between $\mathbf{E}t$ and the lattice needs to be computed in a non-trivial $D$-dimensional geometry whose metric is defined by the $Q$-matrix (27). The problem of finding the point on a regular lattice that is closest to a real vector in a given lattice

---

[6]We note however that in very special cases where the integrable structures are very constrained, the actual minimizing geodesics can be of the form of our Ansatz [5,6]. In this context, the numerical tools we shall describe below were able to find the right geodesics and our upper bound in fact saturates Nielsen's complexity.

[7]In fact, at very large $\mu$ the gap between the variational upper bound and the true complexity will be large. In the limit $\mu \to \infty$, the geometry becomes sub-Riemannian and the geodesics are piecewise smooth curves, where each segment is a geodesic whose velocity is purely local. This result is known as the Chow-Rashevskii theorem [25] (see also [21, 22]). Nielsen's complexity therefore converges towards a finite value when $\mu \to \infty$, whereas our bound grows without limit with $\mu$. The best strategy to use our bound is therefore to identify a value of $\mu$ beyond which Nielsen's complexity essentially stops growing, and apply our bound at that value of $\mu$.

geometry is known as the *closest vector problem* (CVP), and is often discussed in lattice-based cryptography [26]. We find that the $Q$-matrix is a key ingredient of the upper bound (26), since its properties fully determine the geometry of the lattice on which the optimization needs to be performed. Such $Q$-matrices defined through energy eigenvectors of physical systems will therefore be the central object of our studies.

### 2.1.4 Numerical algorithms to approximately solve the CVP

Despite the conceptual simplicity of the final geometric problem, it remains extremely difficult to find an exact solution to (26), especially at large $D$. However, to investigate whether different types of dynamics are distinguished with the variational Ansatz it may be sufficient to only approximately solve (26). Fortunately, due to the importance of the CVP in cryptographic protocols, a large body of work has been dedicated to finding good approximate solutions to lattice minimization problems. When choosing which algorithm to implement, a relevant question is whether the output needs to be as accurate as possible, since precision often implies a longer running time. It is an active field of research in lattice optimization to try and improve the efficiency of numerical methods in this area. In [6], several lattice optimization algorithms running in polynomial time were found to be useful in relation to our current perspective. We shall review them briefly below as these will be our main tools in the analysis of section 4. Although the theoretical precision of these algorithms is often less impressive than approaches for which the running time scales exponentially with $D$, the numerical methods we will now discuss have been shown to usually perform considerably better in practice than the proved worst performance bounds [27]. They furthermore suffice for constructing practical estimates of Nielsen's complexity that distinguish integrable and chaotic dynamics.

*The Lenstra-Lenstra-Lovász algorithm*

The main obstruction to solving (26) in a way analogous to (19) is the deformation of the lattice geometry with the $Q$-matrix. Indeed, to obtain (19) in the Euclidean geometry corresponding to the bi-invariant complexity, we relied on the fact that the standard $D$-dimensional integer lattice basis vectors are mutually orthogonal with respect to the Euclidean metric. This allowed us to project the $n$-th component of the vector $\mathbf{E}t/2\pi$ to the nearest integer in the $n$-th (standard) direction of the lattice. This orthogonality property is no longer valid in a generic metric $\mathbf{I} + (\mu-1)\mathbf{Q}$. To get some more intuition, one can consider a dual picture obtained by performing a linear transformation on the lattice to bring the metric to the Euclidean form [6]. As a result of this rotation, the standard unit cell could turn out to be very elongated and not defining an optimal basis to find the closest lattice vector to a real vector projection onto hypersurfaces of the lattice. Note that any integer combination of the basis vectors which generates the entire lattice would do as a valid basis and, given a basis with bad orthogonality properties, there most certainly exists a modified lattice basis that is closer to being orthogonal. This is precisely what the Lenstra-Lenstra-Lovász (LLL) algorithm [28] seeks to find. The reduced basis generally consists of shorter vectors that are consequently more orthogonal.[8] This algorithm for improving the lattice basis is a key tool for approximately solving the CVP, and hence finding useful bounds on Nielsen's complexity.

*Babai's nearest plane algorithm*

Even after having performed a basis reduction using the LLL-algorithm, the resulting lattice basis is generically still non-orthogonal. As a consequence, there is no guarantee that the process of naively rounding the coefficients of the real vector expanded in the LLL-reduced

---

[8]Note that the volume of the unit cell is independent on the choice of basis.

lattice basis produces a good solution to the CVP. The main shortcoming of this method is that it projects the $D$ components of the vector independently, although pairs of basis vectors may have significant overlap. To remedy this, Babai devised his nearest plane algorithm [29]. The method starts by partitioning the original lattice in an infinite number of $(D-1)$-dimensional hyperplanes generated by the first $D-1$ lattice basis vectors. By projecting the real vector $\mathbf{E}t$ down to the closest hyperplane, one finds a new real vector which is embedded in a $(D-1)$-dimensional sublattice within the corresponding hyperplane. This defines a CVP in one lower dimension and allows one to recursively reduce the dimension of the CVP until the vector is projected down to a 0-dimensional lattice, which consists of a single point of the original $D$-dimensional lattice. This method gradually projects the coefficient of the basis vectors to integers by taking previous projections into account in the minimization algorithm and results in an estimate for the lattice vector closest to $\mathbf{E}t$ in the metric $\mathbf{I} + (\mu-1)\mathbf{Q}$. It is known to typically perform better than naive rounding of the vector components, and has been useful in our context.

*The greedy algorithm*

Finally, we apply a last optimization algorithm to the lattice vector constructed with the combined methods of LLL and Babai, which is based on the 'greedy' algorithm [30]. Given an approximate solution to the CVP and an LLL-reduced basis, the greedy search verifies whether moving along a lattice direction $b_i$ in integer steps $c_i$ improves the solution. The approximate solution is then updated by a subtraction $c_i b_i$ in the direction and magnitude for which the gain is maximized. The greedy algorithm continues until no such subtractions can improve the minimization process. The output of this last algorithm then constitutes our best guess at a solution to (26).

A detailed discussion of these three algorithms together with a summary of their estimated performance can be found in [6]. Interestingly, the final saturation value of the complexity curve resulting from a late-time approximate CVP solution was found to be reasonably well estimated using properties of the LLL-reduced lattice basis alone [6]. The argument to obtain this estimate is based on the following picture. At late times, the vector $\mathbf{E}t$ is expected to be a typical point in $\mathbb{R}^D$. It turns out that the distance of a typical point to the lattice can be estimated by means of a quantity known as the covering radius $\rho$. The covering radius is defined as the minimal radius of identical $D$-dimensional spheres, centered at every point of the lattice, such that this collection of spheres would cover the entire $D$-dimensional space. In terms of the covering radius, the average distance of a real point in $\mathbb{R}^D$ to a lattice has been conjectured to be at most $\rho/\sqrt{3}$ [31]. Naturally, finding the covering radius in any high-dimensional lattice is hard. This quantity can nevertheless be shown to be upper bounded by a simple expression involving the orthogonal set of vectors $\{\mathbf{b}_i^*\}$ obtained after applying the Gram-Schmidt orthogonalization procedure to an arbitrary lattice basis [26]

$$\rho(\mathcal{L}) \leq \frac{1}{2}\left(\sum_i ||b_i^*||^2\right)^{1/2} . \tag{28}$$

This upper bound is most effective when considered on a good lattice basis, as also observed in Figure 11 of [6]. A good basis aims to find short and mutually orthogonal lattice vectors, which puts constraints on the length of the corresponding Gram-Schmidt vectors appearing in (28). In [6], the RHS of (28) evaluated for the LLL-reduced basis was taken as a rough estimate for $\rho$. This, in turn, lead to an educated guess for the late-time value of $\mathcal{C}_{\text{bound}}(t)$

$$\mathcal{C}_{\text{average estimate}} = \frac{\pi}{\sqrt{3}}\left(\sum_i ||b_i^*||^2\right)^{1/2} , \tag{29}$$

where the additional factor of $2\pi$ originates from the unnormalized lattice spacing in $(2\pi\mathbb{Z})^D$. This was found to be a successful approximation in practice, for both integrable and chaotic models (as shown in Figure 10 of [6]).

It is worthwhile to remark that, while the energy eigenvalues are the only ingredients in the evolution of the bi-invariant complexity, the eigenvectors have a predominant role when $\mu \neq 1$ as they determine the geometry of the lattice. As we now discuss, valuable information about the integrable properties of a quantum system is encoded in the $Q$-matrix.

### 2.1.5 Properties of the *Q*-matrix

Sections 3 and 4 will be focused on the interplay between integrable features of quantum dynamics and complexity reduction, through the geometric picture suggested by (26). The characteristics of the $Q$-matrix in chaotic and integrable models shall be a central object in this analysis. Let us therefore start by collecting some of the elementary properties of the $Q$-matrix that were initially derived in [6]. It is straightforward to show that the $Q$-matrix is non-negative with eigenvalues between 0 and 1. When the locality degree $k$ is low compared to the number of degrees of freedom, most of the generators are non-local. As a consequence, the number of elements in the sum over $\alpha$ in (27) is small and the $Q$-matrix is close to the identity, with a distribution of eigenvalues that peaks near 1. Increasing $k$ allows more generators into the set of local operators and pushes the mean and peak of the $Q$-matrix eigenvalue distribution to smaller values. Moreover, it was observed in [6] that the fashion in which the distribution is deformed from 1 towards the origin as $k$ increases depends on the dynamical features of the model.

An important part of the data encoded in the $Q$-matrix lies in its kernel, which contains rich information about the locality structure of potential conservation laws of the system. Consider a vector $\{c_n\}$ pointing along a null direction of the $Q$-matrix, and write

$$0 = \sum_{mn} Q_{mn} c_m c_n = \sum_{\dot\alpha} \left( \sum_n \langle n | T_{\dot\alpha} | n \rangle c_n \right)^2 = \sum_{\dot\alpha} \left( \mathrm{Tr}\left[ T_{\dot\alpha} \sum_n c_n | n \rangle \langle n | \right] \right)^2 . \quad (30)$$

Since every term in the right-hand-side of (30) needs to vanish independently, any null direction produces an operator that is diagonal in the Hamiltonian eigenbasis and which has no overlap with any of the non-local generators. As a result, there is a one-to-one correspondence between *local* linearly independent conservation laws of a quantum system (i.e., the conservation laws entirely constructed of the 'easy' operators used to define Nielsen's complexity)

$$\mathcal{O} \equiv \sum_n c_n | n \rangle \langle n | , \qquad [H, \mathcal{O}] = 0 , \qquad \sum_{\dot\alpha} \mathrm{Tr}[T_{\dot\alpha} \mathcal{O}] = 0 , \quad (31)$$

and the null eigenvalues of the corresponding $Q$-matrix. Remarkably, therefore, the $Q$-matrix emerges as a useful tool which provides us with an algorithmic procedure to find conservation laws with local properties in systems where integrable features are suspected. Note that, in the same manner, and using the second expression for the $Q$-matrix (27), conserved quantities that can be expanded purely in non-local directions define an eigenvector of the $Q$-matrix with eigenvalue equal to 1.

Null directions (30) are interesting in that they allow for directions in the CVP lattice which do not suffer from the penalty $\mu$. This has the potential to lead to a drastic reduction of the complexity (26) compared to systems with no vanishing $Q$-eigenvalues. As mentioned above, the locality degree is generally chosen such that the Hamiltonian is local. The $Q$-matrix of physical systems is therefore assumed to always have at least one null direction.[9]

---

[9]Moreover, if all the non-local operators are chosen to be traceless (i.e., if the identity operator belongs to the

The observed connection between null vectors of the $Q$-matrix and local conserved operators is a clear indication that the spectrum of the $Q$-matrix contains non-trivial information about the integrable structures of a model. The $Q$-eigenvalue distribution can hence be expected to be quite different for chaotic and integrable models. It is generally not straightforward to define integrability in finite-dimensional quantum models precisely. Indeed, any $D \times D$ Hamiltonian, chaotic or integrable, commutes by elementary linear algebra with $D - 1$ linearly independent matrices. It is therefore legitimate to ask what distinguishes the set of commuting matrices of chaotic and integrable Hamiltonian. The point is that the organization of conservation laws in integrable models often involves a tower of conserved quantities of definite, increasing locality degree (e.g. quantum systems which become Lax integrable in the classical limit). This suggests in particular that the number of null eigenvalues of the $Q$-matrix should steadily increase for integrable models when cranking up the locality degree $k$ while keeping other parameters, such as the number of particles, fixed.

### 2.1.6 Generic features of the upper bound on complexity

It is widely believed that integrable evolution is fundamentally less complex than chaotic dynamics, and that this should be reflected in a manifest complexity reduction for integrable models. Yet, there are few demonstrations of such behaviors using concrete complexity measures, nor is there much certainty in how much complexity reduction one may legitimately expect. In [6], we introduced and studied in a few concrete models the time dependence of the upper bound on Nielsen's complexity (26). Applied to bosonic systems [32] and fermionic systems [5], with or without various types of integrable structures, this analysis resulted in the following observations. First, the complexity curves consistently display an initial linear growth, as in (22), and a late-time saturation region. This behavior is widely believed to be generic for complexity measures [4, 5, 33]. Second, the variational Ansatz was found to successfully identify integrable models as less complex than chaotic models. This has been manifest in two different regions of the complexity curve. In some integrable models, sharp downwards pointing spikes appear during the initial linear growth. More systematically, however, the distinction between the two types of dynamics was observed in the late-time saturation region. As one increases the locality threshold $k^{max}$, a gap emerges between the complexity plateau heights of integrable and chaotic models. The complexity saturation heights for both types of dynamics were found to scale with $\sqrt{\mu D}$, with a lower prefactor for the integrable instances.

These earlier results motivate the search for a deeper understanding of the connection between integrability and complexity reduction. Spin chain models appear to be ideal candidates for this task. Their integrable properties have been extensively studied and their spatial extent offers a new perspective to test the robustness of the conclusions of [6]. The relation between integrability and complexity reduction in spin chains shall be the focus of section 4. For chaotic models, on the other hand, we will argue in section 3 that the shape of the $Q$-matrix eigenvalue distribution is essentially universal in the large $D$ limit, and shares many of the features of 'artificial' $Q$-matrices emerging from random basis vectors. This leads to a universal prediction for the saturation height of the complexity bound for chaotic models. First, however, we need to extend the methods of [6] to degenerate energy spectra, which are generically displayed by spin chains.

---

set of local directions), the all-one vector defines an additional null vector of the $Q$-matrix. While this choice was adopted in [6], it will be more convenient in the following to declare the identity to be non-local. This is straightforward to impose since spin chain Hamiltonians are usually traceless. In this case, the all-one vector is an eigenvector of the $Q$-matrix with eigenvalue 1.

## 2.2 Extension to degenerate energy spectra

In the absence of degenerate levels in the energy spectrum, a solution to the CVP (26) provides an exact minimum of the variational Ansatz on Nielsen's complexity. This picture is no longer true for a degenerate Hamiltonian, since there is an inherent ambiguity in the choice of energy eigenbasis in (26)-(27). There is a freedom in constructing operators of the form $e^{-iVt}$ with boundary condition (3) using a variety of velocities (17) expanded in different bases for the degenerate subspaces.

Finding the shortest bi-invariant geodesic which minimizes the variational Ansatz on Nielsen's complexity will therefore also involve a search for the optimal energy eigenbasis in which to perform the CVP (26). There are multiple ways one could approach this complex minimization procedure. Perhaps most straightforwardly, one could parameterize the enlarged set of solutions to (17) by introducing a set of angles for each degenerate subspace which effectively rotate different eigenbases of that subspace into each other. This approach however greatly increases the difficulty of solving the variational problem, since one is then required to optimize (26) over the discrete set of integers $k_n$ as well as over a (potentially large) number of continuous parameters. While any given choice of energy eigenbasis leads to a geometric picture similar to (26), the introduction of continuous angles parametrizing the eigenbasis of $H$ leads to an angle-dependent $Q$-matrix defining the lattice geometry. From our discussion above, it is natural to expect that the resulting complexity evolution may be the lowest for a choice of angles that maximizes the number of small $Q$-eigenvalues. In the assumption that, in the large $D$ limit, the distribution of eigenvalues of the $Q$-matrix for generic $k$ approaches a smooth curve, the number of small eigenvalues should correlate with the dimension of its kernel. Using this intuition, we propose a method consisting in finding an energy eigenbasis with the largest number of null directions for (27) as a first step in setting up a variational Ansatz for degenerate Hamiltonians. This eigenbasis shall subsequently be used in (26), which leads to a conventional CVP. Recall that the number of zero $Q$-eigenvalues is in general dependent on the locality threshold $k$, which may require selecting a (different) appropriate basis at every locality threshold.

Note that, for the $Q$-matrix to have the largest number of zeros, the eigenbasis of the Hamiltonian should be aligned with the eigenbases of other conserved quantities. For chaotic systems, there will typically be no preferred energy eigenbasis since we do not expect any other local conservation law besides the Hamiltonian. By contrast, integrable models include a tower of conserved charges which are in involution and moreover do not generally share the degeneracies of the Hamiltonian. The requirement that the set of local conservation laws should be diagonal in the chosen eigenbasis is therefore often enough to fully specify a unique basis of eigenvectors for $H$. However, this avenue requires at least partial knowledge of the tower of conservation laws, which is not always accessible.

One can nevertheless make progress without any prior knowledge of the integrable properties of the system, as we shall now describe. Consider an arbitrary basis of eigenvectors $|n_i\rangle$ for each degenerate eigenspace of the Hamiltonian, where $n$ labels the energy eigenvalue $E_n$ and $i$ the eigenvectors belonging to the associated degenerate subspace of size $d_n$. To find a preferred eigenbasis which maximizes the number of zero eigenvalues of the $Q$-matrix, let us start by examining an 'enlarged' $\tilde{Q}$-matrix:

$$\tilde{Q}_{IJ} \equiv \sum_{\dot{\alpha}} \langle I_1|T_{\dot{\alpha}}|I_2\rangle \langle J_1|T_{\dot{\alpha}}|J_2\rangle = \delta_{IJ} - \sum_{\alpha} \langle I_1|T_{\alpha}|I_2\rangle \langle J_1|T_{\alpha}|J_2\rangle, \qquad (32)$$

where the indices $I$ and $J$ run, within each degenerate eigensubspace, over all possible pairwise combinations $I = (I_1, I_2) = (n_i, n_j)$ and $J = (J_1, J_2) = (m_i, m_j)$ of the eigenstates $\{|n_i\rangle|i = 1, \cdots, d_n\}$ and $\{|m_i\rangle|i = 1, \cdots, d_m\}$, respectively. (If a level is non-degenerate, it contributes only one value in the list of possible values of $I$ and $J$. If all levels are non-degenerate,

this agrees with the definition of the $Q$-matrix in (27) for non-degenerate spectra.) In analogy with (30), every null vector $\{c_I\}$ of the $\tilde{Q}$-matrix now defines an operator

$$\mathcal{O} = \sum_I c_I |I_2\rangle\langle I_1|, \tag{33}$$

which has no support on the non-local directions $T_{\dot{\alpha}}$ and is block-diagonal with respect to the degenerate energy eigensubspaces. Each element in the kernel of the $\tilde{Q}$-matrix hence defines a local operator that commutes with the Hamiltonian.

Note that this enlarged $\tilde{Q}$-matrix cannot be used in (26) since it does not have the correct dimensions. However, it can be exploited to determine the local conservation laws of $H$ irrespective of the energy eigenbasis we started with. This observation opens up a way to identify the preferred eigenbasis for the degenerate Hamiltonian that maximizes the number of null directions for the $Q$-matrix (27) at a given locality threshold $k$, as follows. First, one computes the enlarged $\tilde{Q}$-matrix (32). From its kernel, one can deduce a complete basis for the local conserved quantities of the system.[10] Then, one performs a simultaneous (exact) diagonalization of the Hamiltonian together with a maximal[11] set of commuting Hermitian combinations of the conserved operators constructed from the null directions of $\tilde{Q}$. By construction, the resulting eigenbasis guarantees to diagonalize the largest possible set of local conservation laws of the Hamiltonian, as required. The $Q$-matrix associated to this new basis via (27) has the largest possible number of null vectors, and can be used to set up a CVP according to (26).

To conclude, we note that with the $\tilde{Q}$-matrix at hand, the final $Q$-matrix does not need to be computed through (27) as this calculation can be numerically time consuming when the number of local generators $T_{\alpha}$ becomes large. Instead, the matrix elements of the enlarged $\tilde{Q}$-matrix contain all the information to construct the $Q$-matrix for any energy eigenbasis. This is easily seen by considering the expansion of the final energy eigenbasis $|n_i'\rangle$ in terms of the original eigenbasis $|n_k\rangle$ used in (32)

$$|n_i'\rangle = \sum_{k=1}^{d_n} a_i^k |n_k\rangle. \tag{34}$$

Then, by evaluating (27) for (34)

$$Q_{n_i'm_j'} \equiv \sum_{\dot{\alpha}} \langle n_i'|T_{\dot{\alpha}}|n_i'\rangle\langle m_j'|T_{\dot{\alpha}}|m_j'\rangle \tag{35}$$

$$= \sum_{k,k'=1}^{d_n} \sum_{l,l'=1}^{d_m} \left(a_i^k b_j^l\right)^* a_i^{k'} b_j^{l'} \left(\sum_{\dot{\alpha}} \langle n_k|T_{\dot{\alpha}}|n_{k'}\rangle\langle m_l|T_{\dot{\alpha}}|m_{l'}\rangle\right), \tag{36}$$

one observes that the terms that appear in this expansion are proportional to the matrix elements of (32).

The procedure to find a preferred energy basis for a degenerate energy spectrum that has just been outlined is general and does not require any information about the integrable structure of a quantum system. As mentioned above, however, when the degeneracies originate from known unitary symmetries, one can straightforwardly diagonalize the Hamiltonian

---

[10]We note that, although the dimension of the matrix $\tilde{Q}$ can be much larger than the dimension of the Hamiltonian in the presence of large degenerate subspaces, one is only interested in the directions corresponding to the lowest lying eigenvalues of $\tilde{Q}$. These can be found either by exact diagonalization or any numerical algorithm that is suitable to find all the null directions of $\tilde{Q}$ efficiently (such as e.g. the Lanczos algorithm).

[11]In practice, we find that the conserved operators identified in this way are all in involution. This is not surprising since the integrable towers are in involution, and usually also respect the subgroup of global symmetries that lead to additional local conservation laws.

simultaneously with a set of mutually commuting conserved quantities. This should automatically select a good energy eigenbasis to feed into (27), but has the drawback that it requires complete knowledge of the conserved quantities of the model at a given locality threshold $k$ beforehand.

When a system possesses multiple global symmetries the sizes of the degenerate subspaces can quickly grow. This can result in a huge enlarged $\tilde{Q}$-matrix that is hard to handle numerically. It can therefore also be wise to pursue a hybrid strategy and use a small number of global symmetries to split some of the degeneracies prior to constructing the $\tilde{Q}$-matrix, in order to reduce its size. However, this only works if the resulting local conservation laws commute with the global symmetries used in the construction. In practice, we determine the right combinations of global symmetries for small chains by trial and error.

## 3 Random matrix theory of complexity saturation in chaotic systems

In this section, we shall argue that the difference in late-time saturation values of the bound on complexity (26) between chaotic and integrable systems can be traced back to the properties of the $Q$-matrix. In the case of integrable models, the change of the size of the kernel of the $Q$-matrix as the locality parameter $k^{max}$ is increased was observed to have a direct influence on the amount of small but non-zero eigenvalues of the $Q$-matrix [6]. A relation between zero and small eigenvalues is not implausible as one can suppose that the eigenvalue distribution of the $Q$-matrix tends to a continuous curve in the large $D$ limit in generic systems. Therefore, a small increase in the number of zero eigenvalues can lead to a fair amount of small eigenvalues. Geometrically, this defines many directions on the manifold of unitaries which a curve can explore at lower costs. Moreover, the flattened shape of the $Q$-eigenvalue distributions at larger $k^{max}$ was found to correlate with corresponding complexity reductions observed in integrable systems. In contrast, the $Q$-matrix eigenvalue distributions associated to chaotic Hamiltonians display very few low eigenvalues at moderate values of $k^{max}$ and appeared as a generic bell-shaped curve peaked about a value that evolves from 1 to 0 when dialing the locality threshold $k^{max}$ from small to large values. In fact, the bound on complexity obtained for chaotic models by means of the three algorithms outlined above was found to be very similar to crude bounds resulting from less sophisticated methods, such as a naive rounding approach to finding **k** in (26) [6]. This observation suggests that the lattice geometry induced by the $Q$-matrix for chaotic Hamiltonians might in fact already be quite close to Euclidean and that there is little to no advantage of using refined methods to estimate their complexity, in contrast to the case of integrable models. We shall verify and investigate the consequences of this statement in the present section.

The common lore of quantum chaos [36–38] suggests that some features of generic chaotic models are universal and well-captured by random matrix theory (RMT). In section 3.1, we show that one can gain insight in the shape of the $Q$-matrix eigenvalue distribution for generic chaotic models by relying on the statistical properties of random unitary matrices. In particular, we aim to show that the eigenvalues of typical $Q$-matrices concentrate around their mean value at large $D$. Assuming such concentration, we show in section 3.2 that the first moment of the distribution of the $Q$-eigenvalues fixes the saturation height of the complexity curve of chaotic models. In the next section, we will discuss how these theoretical estimates compare against the numerical moments of $Q$-matrices of spin chains.

## 3.1 Distribution of $Q$-eigenvalues for chaotic models

One implication of the connection between RMT and chaotic quantum systems is that the components of the eigenvectors of the Hamiltonian expressed in any non-fine-tuned basis are essentially random variables (see e.g. [39, 40] for reviews). We shall be interested in computing the first two moments of the $Q$-eigenvalue distribution associated to a Hamiltonian whose eigenvectors can be viewed as the columns of a random unitary matrix drawn from the Gaussian Unitary Ensemble (GUE).[12] We denote the eigenvalues of the $Q$-matrix $\lambda_n$ and define $\lambda = \{\lambda_1, \ldots, \lambda_D\}$. Random matrix theory can then be used to produce an estimate for the mean value

$$\bar{\lambda} \equiv \frac{1}{D}\text{Tr}(Q) = \frac{1}{D}\sum_n Q_{nn}, \tag{37}$$

as well as the variance

$$\text{Var}(\lambda) \equiv \frac{1}{D}\text{Tr}(Q^2) - \frac{1}{D^2}\text{Tr}(Q)^2 = \frac{1}{D}\sum_{n,m}Q_{nm}Q_{mn} - \frac{1}{D^2}\left(\sum_n Q_{nn}\right)^2, \tag{38}$$

of the spectrum of a typical chaotic $Q$-matrix, in the large $D$ limit. For simplicity, we assume that the identity is not part of the set of local generators. This allows us to systematically neglect traces of single local generators. Occasionally, we comment on the effects of including the identity in the discussion.

### 3.1.1 RMT prediction for the mean

We start by expanding the eigenvectors of the Hamiltonian $|n\rangle$ in (27) in a fixed, non-fine-tuned basis $|i\rangle$. This basis can be assumed to be a standard physically motivated basis (like the Fock basis, or the single-site spin basis), though the details will not be relevant. Defining the overlap $\psi_i^n \equiv \langle i|n\rangle$, one can write the mean (37) as

$$\bar{\lambda} = 1 - \frac{1}{D}\sum_{n,i,j,k,l=1}^{D}\sum_{\alpha=1}^{N_{loc}}(T_\alpha)_{ij}(T_\alpha)_{kl}(\psi_i^n)^*\psi_j^n(\psi_k^n)^*\psi_l^n, \tag{39}$$

where we used the general expression for the $Q$-matrix (27), in terms of the local generators. The average value of the mean (39), which we denote by $\langle\bar{\lambda}\rangle$, is then determined by a four-point function of the matrix elements of a unitary matrix $\psi$ drawn from GUE.

Moments of the unitary group, equipped with the Haar probability measure, can be expressed in terms of Weingarten functions [41] (see [42] for a summary), which reflect the combinatorial structure of the expectation values. In general, an expectation value of the form

$$\langle(\psi_{i_1}^{n_1})^*\ldots(\psi_{i_d}^{n_d})^*\psi_{j_1}^{m_1}\ldots\psi_{j_{d'}}^{m_{d'}}\rangle \equiv \int_{U(D)}(\psi_{i_1}^{n_1})^*\ldots(\psi_{i_d}^{n_d})^*\psi_{j_1}^{m_1}\ldots\psi_{j_d'}^{m_d'}d\psi, \tag{40}$$

vanishes unless $d = d'$. This fact follows straightforwardly from the invariance of the Haar measure under multiplication by a complex number of norm 1.

---

[12]For concreteness, we choose to evaluate these estimates in the GUE. However, in the large $D$ limit all statistical ensembles give a similar scaling with $D$, although the numerical coefficients might differ. The general features of the $Q$-eigenvalue distribution we seek to describe, namely that the distribution concentrates at large $D$ and peaks near 1 in the thermodynamic limit are common to all standard random matrix ensembles. The spin chains that will be considered in section 4 are time-reversal symmetric. The spectral statistics of the chaotic instances are therefore expected to follow the GOE Wigner-Dyson distribution. This means that there exists a real Hamiltonian eigenbasis for these models. However, the eigenbasis that will be relevant in the numerics is generally complex. This follows from the fact that it is constructed as a common eigenbasis of the Hamiltonian and the momentum operator (58) and this latter operator is not time-reversal symmetric.

When $d = d'$, the $2d$-point function of unitary matrix elements is constrained to take the form [41]

$$\langle (\psi_{i_1}^{n_1})^* \dots (\psi_{i_d}^{n_d})^* \psi_{j_1}^{m_1} \dots \psi_{j_d}^{m_d} \rangle = \sum_{\sigma, \tau \in S_d} \delta_{i_1 j_{\sigma(1)}} \dots \delta_{i_d j_{\sigma(d)}} \delta_{n_1 m_{\tau(1)}} \dots \delta_{n_d m_{\tau(d)}} \mathrm{Wg}^U(\tau \sigma^{-1}, D),$$
(41)

where the two sums run over the permutation group of $d$ elements. The unitary Weingarten function $\mathrm{Wg}^U(\cdot, D)$ of a permutation is fully determined by the cycle structure of its permutation argument and corresponds to an explicit function of $D$. The relevant expressions in the present context can be found in Appendix A. The structure of (41) tells us that a generic expectation value is determined by all possible pairings of equal indices (lower and upper separately) of the conjugated and non-conjugated matrix elements. In addition, each of the contribution is weighted by a function of the dimension $D$ that is determined by the permutation involved in the pairings of the two sets of indices.

Applying this formula in the context of (39), the expectation value of the four-point function appearing in (39) requires us to consider a sum over all possible permutations $\sigma$ and $\tau$ of the index sequences $(i, k)$ and $(n, n)$, respectively, and compute the scaling in $D$ associated to each term. Since all the upper indices are equal, half of the Kronecker delta symbols collapse. Collecting identical pairings for the remaining indices leads to

$$\langle (\psi_i^n)^* (\psi_k^n)^* \psi_j^n \psi_l^n \rangle = \frac{1}{D(D+1)} (\delta_{ij} \delta_{kl} + \delta_{il} \delta_{kj}).$$
(42)

We then estimate the average value of $\bar{\lambda}$ (39) using (42) and find

$$\langle \bar{\lambda} \rangle = 1 - \sum_{\alpha=1}^{N_{loc}} \frac{1}{D(D+1)} \left( (\mathrm{Tr}[T_\alpha])^2 + \mathrm{Tr}[T_\alpha^2] \right).$$
(43)

In the large $D$ limit, taking the tracelessness and the normalization (4) of the generators into account, the RMT estimate for the mean of the $Q$-eigenvalue distribution for chaotic models reduces to

$$\langle \bar{\lambda} \rangle = 1 - \frac{N_{loc}}{D^2}.$$
(44)

In most models, it is natural to expect the number of local operators to grow with the Hilbert space dimension as $(\log D)^p$, for some power $p$ that depends on the locality threshold $k$. Hence, in the thermodynamic limit where $D$ goes to infinity while keeping $k$ fixed, the $Q$-eigenvalue distribution of generic chaotic models is predicted by RMT to cluster near 1.

When the (appropriately normalized) identity operator is included as an element of the set of local generators in (43), it contributes an additional $-1/D$ to the estimate (44), at large $D$.

### 3.1.2 RMT prediction for the variance

We proceed with the computation of the variance on the distribution of $Q$-eigenvalues, (38). Expressing the energy eigenstates in terms of random unitary vectors, one can write

$$\mathrm{Var}(\lambda) = \frac{1}{D} \sum_{\substack{n,m,i,j, \\ k,l,o,p,q,r}} \sum_{\alpha, \beta} (T_\alpha)_{ij} (T_\alpha)_{kl} (T_\beta)_{op} (T_\beta)_{qr}$$

$$\times \left( (\psi_i^n)^* (\psi_k^m)^* (\psi_o^m)^* (\psi_q^n)^* \psi_j^n \psi_l^m \psi_p^m \psi_r^n - \frac{1}{D} (\psi_i^n)^* (\psi_k^n)^* (\psi_o^m)^* (\psi_q^m)^* \psi_j^n \psi_l^n \psi_p^m \psi_r^m \right),$$
(45)

which can be estimated by replacing the products of unitary matrix elements by their expectation values in the GUE. Note that the contribution from the $\delta_{nm}$ in (27) drops out in the final expression for the variance (45).

The derivation of eight-point functions in the GUE is rather cumbersome but can nevertheless be done exactly using (41). Both terms in (45) involve an evaluation of all permutations of the two index sequences (upper and lower) of four indices of the non-conjugated matrix elements. For this, we distinguish between the cases $m = n$ and $m \neq n$. We simplify the computation by working to leading order in both $1/D$ and the ratio

$$r \equiv \frac{N_{loc}}{D^2}. \tag{46}$$

Similar to (43), the products of Kronecker delta symbols appearing in (41) can be reorganized in terms of products of traces of combinations of the four generators through the sums in (45). When all the local operators are traceless, only the following traces contribute to the variance:

$$\mathrm{Tr}[T_\alpha^2]\mathrm{Tr}[T_\beta^2], \qquad \left(\mathrm{Tr}[T_\alpha T_\beta]\right)^2, \qquad \mathrm{Tr}[T_\alpha^2 T_\beta^2], \qquad \mathrm{Tr}[T_\alpha T_\beta T_\alpha T_\beta]. \tag{47}$$

The first two of these expressions are fixed by the normalization of the generators (4) and do not scale with $D$. The contribution from the remaining two is theory-dependent but we shall assume that these are generally suppressed by a factor of $1/D$. In spin $1/2$ chains, the standard generators are strings of Pauli operators, which have the property to square to a multiple of the identity. Moreover, two such generators anti-commute if they share an odd number of sites with distinct, non-trivial Pauli operator and commute otherwise. Therefore, with the normalization (4) one has

$$\mathrm{Tr}[T_\alpha^2 T_\beta^2] = \frac{1}{D}, \qquad \mathrm{Tr}[T_\alpha T_\beta T_\alpha T_\beta] = \pm\frac{1}{D}. \tag{48}$$

In the following, we will assume that this scaling produces an accurate estimate for the contribution of the connected traces in other models as well.

A complete list of index permutations contributing to each term in (45), with associated expressions in terms of traces, and corresponding Weingarten functions can be found in [43],[13] for $n \neq m$ and $n = m$. Since we are interested in the large $D$ behavior, it suffices to consider the leading contributions from each of the two types of terms in (45). We find that these are given by

$$\langle \mathrm{Var}(\lambda) \rangle \approx \frac{1}{D^5} \sum_{\alpha,\beta} \left[ \sum_{n \neq m} \left( \left(\mathrm{Tr}[T_\alpha T_\beta]\right)^2 - \frac{1}{D}\mathrm{Tr}[T_\alpha^2]\mathrm{Tr}[T_\beta^2] \right) + \sum_n \left(1 - \frac{1}{D}\right)\mathrm{Tr}[T_\alpha^2]\mathrm{Tr}[T_\beta^2] \right]$$
$$\sim \frac{N_{loc}}{D^3} \left( 1 + \mathcal{O}\left(\frac{N_{loc}}{D^2}\right) + \mathcal{O}\left(\frac{1}{D}\right) \right). \tag{49}$$

This result provides a theoretical explanation for the observed concentration in the $Q$-eigenvalue distribution in chaotic models. We find that the standard deviation of the $Q$-eigenvalue distribution is set to leading order by $(r/D)^{1/2}$. In the large $D$ limit at fixed locality threshold, (44) and (49) tell us that the mean $\langle \bar{\lambda} \rangle$ approaches 1 faster than the standard deviation approaches 0. Alternatively, one can let $D$ tend to infinity while fixing $r$, in which case the distribution of the spectrum of the $Q$-matrix displays a peak about the constant mean value

---

[13]The data and code used for the numerics in this article are available in the Zenodo data repository at https://doi.org/10.5281/zenodo.7876467.

$\langle \bar{\lambda} \rangle = 1 - r$ with a standard deviation decreasing as $D^{-1/2}$. In the large $D$ limit, the distribution concentrates about the mean.

In the next section, we will examine how well these features predicted by RMT are reflected in physical $Q$-matrices. It is straightforward to verify that the RMT predictions (44) and (49) for the shape of the $Q$-eigenvalue distribution agree quantitatively with the numerically computed mean and variance for randomly generated unitary matrices playing the role of the Hamiltonian eigenbasis, as they should. This can be verified for any choice of easy generators, not necessarily selected on the basis of locality. However, it is well-known that only a selection of features of RMT are expected to be generic enough to be universally observed in chaotic Hamiltonians. For example, the level spacing statistics of chaotic models are expected to be consistent with RMT, while the energy level density is not. In reality, most physical Hamiltonians are very sparse in their standard basis since they usually describe few-body interactions. The resulting energy eigenbasis therefore does not necessarily share all the features of elements drawn from the GUE, which is only expected to be a valid description for many-body interactions at large $k$. It is therefore important to discuss potential discrepancies between these theoretical estimates and genuine physical $Q$-eigenvalue distributions and possibly pinpoint their origin.

Although the choice of local generators $T_\alpha$ did not play any crucial role in the derivation of (44) and (49), it is an essential ingredient when it comes to describing the differences between GUE predictions and physical models. Indeed, when considering genuine Hamiltonian data together with a physically motivated set of generators, the $Q$-matrix always has at least one zero eigenvalue, which corresponds to the Hamiltonian itself. The presence of this null direction inevitably pushes the mean to a lower value and the variance to a higher value (at least by $1/D$). For this reason, the agreement between RMT estimates and the moments of real distributions is expected to be qualitative at best. For generic chaotic models, this null direction is likely to be the only zero eigenvalue of the $Q$-matrix when $k \ll N$, with $N$ a measure for the number of degrees of freedom (e.g. the number of spins) and this small deviation does not qualitatively change the overall pattern of the $Q$-eigenvalue distribution.

As mentioned previously, the dependence of the $Q$-eigenvalue distributions on the locality threshold $k$ for integrable models was observed to be quite distinct from generic models. In particular, the concentration property is absent already at moderately small values of $k$. Indeed, the locality properties of their conserved charges interferes with this picture and puts additional constraints on the spectrum of the $Q$-matrix by requiring a specific scaling of the number of null directions with increasing locality degree $k$. This should be reflected in the overlaps $\psi_i^n$, which in general cannot be well approximated by random vectors.

We note that the variance (49) is distinct from the variance on the distribution of the mean values of many random $Q$-matrix realizations. The quantity (49) is an estimate for the variance of the distribution of the $Q$-eigenvalues for one instance of a random matrix $\psi_i^n$. The variance on the distribution of means $\bar{\lambda}$ obtained individually for each member of set of random matrices $\psi$ is given by

$$\mathrm{Var}(\bar{\lambda}) = \langle \bar{\lambda}^2 \rangle - \langle \bar{\lambda} \rangle^2 \,. \tag{50}$$

The second term can be simply obtained by squaring (44), whereas the first term can be calculated in an analogous fashion to (49) and requires the evaluation of

$$\mathrm{Var}(\bar{\lambda}) = \frac{1}{D^2} \sum_{\substack{n,m,i,j,\\k,l,o,p,q,r}} \sum_{\alpha,\beta} (T_\alpha)_{ij} (T_\alpha)_{kl} (T_\beta)_{op} (T_\beta)_{qr} \langle (\psi_i^n)^* (\psi_k^n)^* (\psi_o^m)^* (\psi_q^m)^* \psi_j^n \psi_l^n \psi_p^m \psi_r^m \rangle$$
$$- \frac{1}{D^2} \Big( \sum_{n,i,j,k,l} \sum_\alpha (T_\alpha)_{ij} (T_\alpha)_{kl} \langle (\psi_i^n)^* \psi_j^n (\psi_k^n)^* \psi_l^n \rangle \Big)^2 \,. \tag{51}$$

A systematic derivation of all terms can be found in [43]. The leading contributions in $1/D$ and $r$ are suppressed compared to (49) and given by

$$\text{Var}(\bar{\lambda}) \sim 2\frac{N_{loc}}{D^5} + 2\frac{N_{loc}^2}{D^6} + \dots \tag{52}$$

The variance on the mean $\bar{\lambda}$ is therefore generally much smaller than the average variance on the $Q$-eigenvalue distribution.

## 3.2 Late-time complexity plateau height from the $Q$-matrix in chaotic models

The concentration property of the distribution of $Q$-eigenvalues for chaotic Hamiltonians provides us with a handle to connect the final average saturation value of the complexity curve to the $Q$-eigenvalues. As we now describe, there exists a direct relation between the plateau height of the complexity curve $\mathcal{C}(t)$ and the mean value of the eigenvalues of the $Q$-matrix.

The idea is that when the eigenvalues of the $Q$-matrix concentrate about their mean value, the resulting matrix can be well approximated by a multiple of the identity. Remember that the $Q$-matrix defines the geometry in which to tackle the CVP (26). The main difficulty of solving CVP in a generic background is that the standard lattice basis vectors are generally not mutually orthogonal with respect to the inner product set by the $Q$-matrix. However, at large $D$ the $Q$-matrix of chaotic models tends to a multiple of the identity, such that the standard lattice basis recovers the properties of a good basis. This provides a potential explanation for the lack of improved performance of the more elaborate methods compared to the naive rounding method observed for chaotic Hamiltonians in [6].

We now use this intuition to explicitly express the complexity plateau height as a function of the first moment of the $Q$-eigenvalues. As discussed above, the averaged late-time complexity can be estimated by the norms of the Gram-Schmidt vectors as in (29). Whenever the $Q$-matrix is well-approximated by the identity, the (LLL-)basis vectors are already approximately orthogonal prior to the Gram-Schmidt orthogonalization and are therefore not substantially altered afterwards. Note that the discussion above (29) as well as the norms appearing in (29) assumed that the lattice metric has been transformed to the Euclidean distance measure by the required rescaling and rotation of the lattice vectors. The Euclidean norms of these rescaled vectors then enter (29). Alternatively, one can simply evaluate these norms by computing the lengths of the standard basis vectors of the lattice $(2\pi\mathbb{Z})^D$ in the metric $\mathbf{I} + (\mu-1)\mathbf{Q}$, taking into account that the factor of $2\pi$ has already been extracted in going from (28) to (29). In summary, (29) can be written in terms of the $Q$-eigenvalues, denoted by $\lambda_n$, as follows

$$\mathcal{C}_{\text{average estimate}} = \frac{\pi}{\sqrt{3}}\left(\sum_{n=1}^{D}(1 + (\mu-1)\lambda_n)\right)^{1/2} \approx \pi\sqrt{\frac{D\mu\bar{\lambda}}{3}}, \tag{53}$$

where we assumed $\mu \gg 1$. We therefore conclude that when the eigenvalues of a $Q$-matrix concentrate around their mean value $\bar{\lambda}$, the saturation height of the complexity curve is controlled by the first moment of the $Q$-eigenvalue distribution. If, in addition, the RMT prediction $\langle\bar{\lambda}\rangle$ for the mean $\bar{\lambda}$ given by (44) is accurate, the plateau height is fixed entirely by the number of local operators $N_{loc}$ and the Hilbert space dimension $D$.

In section 4, we will verify the proposition (53) in detail and show that there is a good qualitative match with numerically computed plateau values for chaotic spin chain models, where concentration of the $Q$-eigenvalues is observed.

# 4 Complexity reduction in integrable quantum spin chains

A central assumption in deriving the estimates (44), (49) and (53) is the absence of any apparent structure in the eigenstates of the chaotic Hamiltonians. When this is a valid assumption, we argued that the late-time solution to the variational Ansatz is essentially fixed by the number of generators unpenalized in the definition of Nielsen's complexity. This conclusion is independent of the specifics of the chaotic dynamics. In contrast, physically adequate complexity measures are expected to display a reduced saturation height for integrable models, a property which has been observed for the upper bound on complexity developed in [6]. From our perspective in section 3, it therefore seems natural to try and understand the origin of this suppression and, in particular, how the properties of integrable Hamiltonians and their eigenstates modify the picture sketched above for chaotic systems.

The results of [6] were suggestive of a correlation between the size of the kernel of the $Q$-matrix (27), which is a direct proxy for the number of independent local conservation laws through (30), and complexity reduction. Null eigenvalues of the $Q$-matrix define directions in the lattice minimization (26) that are insensitive to the large penalty $\mu$. Our working hypothesis is that it is the emergence of these special directions, which effectively constitutes shortcuts on the manifold of unitaries, that permits efficient complexity reduction for integrable models. This mechanism directly relates the locality properties of the integrable structures of a model to its complexity reduction. We note that this has the potential to be relevant to the mechanisms underlying complexity reduction in integrable systems beyond the upper bound (26), since each curve (16) flows to a unique[14] geodesic of the right-invariant metric (10) [3]. In this last section, we verify this hypothesis in detail by an appeal to spin chains. These models provide a fruitful setting for our goals, since there is a wide range of well-studied integrable realizations, which can furthermore easily be perturbed away from integrability. Moreover, unlike the bosonic and fermionic systems studied in [6] which exhibit all-to-all couplings, spin chain interactions respect a notion of spatial locality. This property will allow us to experiment with distinct notions of locality (that is, different recipes for splitting the generators into the 'easy' and 'hard' sets), observe how different notions of locality lead to different amounts of complexity reduction, and see how this reduction correlates with the towers of conserved charges present in integrable systems and their locality properties.

## 4.1 Spin 1/2 chains

We consider a chain consisting of $L$ sites carrying a spin $s$ representation of $SU(2)$ at each site, with nearest neighbor interactions and periodic boundary conditions. We will first focus on the case $s = 1/2$. The single-site spin operators are the Pauli matrices $S_x$, $S_y$ and $S_z$, satisfying the commutation relations

$$[S_a, S_b] = 2i\,\epsilon^{abc}S_c\,. \tag{54}$$

A basis of operators for $SU(2^L)$ is constructed from operators of the form

$$S_{a_1}^{(j_1)}S_{a_2}^{(j_2)}\ldots S_{a_n}^{(j_n)}\,, \tag{55}$$

where $S_a^{(j)}$ represents the $a^{\text{th}}$ Pauli matrix acting on site $j$ and where we assume $j_1 < j_2 < \ldots < j_n$.

Due to the spatial extent in spin chains, one can devise multiple characterizations for the locality of an operator. In the spin 1/2 case, one can put a constraint on the spatial extent of an operator, as well as on the total number of sites where the operators act non-trivially.

---

[14]The deformation procedure from a bi-invariant geodesic to the corresponding right-invariant geodesic is well-defined and produces a unique result as long as no conjugate point is encountered when increasing the cost factor from 1 to the desired value [3].

We denote the parameter specifying the latter notion of locality as $k_{op}$. The spatial locality degree $k_{sp}$ of an operator (55) is defined as the size of the minimal connected 'region' where the operator acts non-trivially, taking the periodic boundary conditions into account. More generally, the locality degree of a linear combination of generators is set by the maximal value of $k_{op}$ and $k_{sp}$ encountered among the individual terms. These maximal values need not be found in the same term.

Splitting the generators into easy and hard operations therefore requires fixing a threshold for the two parameters $(k_{op}, k_{sp})$. Note that for any given operator $k_{op} \leq k_{sp}$. There are a priori at least two straightforward choices of locality threshold. In the context of quantum circuits, it is common to refer to two-qubit gates as operators which act on at most two qubits, without further restrictions. This corresponds to $(k_{op}, k_{sp}) = (2, L)$. One option is hence to set $k_{sp}$ to be the length of the chain, and vary the locality threshold by gradually increasing $k \equiv k_{op}$. For spin chain systems whose dynamical evolution is governed by a local Hamiltonian, on the other hand, it appears more natural to choose a notion of locality based on the spatial structure of the interaction terms. The Hamiltonians we shall consider are built out of nearest neighbor interactions with locality $(k_{op}, k_{sp}) = (2, 2)$. This suggests a second type of locality parametrizing the generators, which can be grouped according to $(k_{op}, k_{sp}) = (k, k)$. Note that this second definition is more restrictive. In particular, the set of local operators as defined by the first characterization contains all the operators that are local according to the second definition at equal parameters.

We can determine the leading order scaling of the number of local operators $N_{loc}$ for large chains, with $k_{sp} = L$ and $k_{op}$ a free parameter. This number is easily found by noting that the number of generators (55) at each value of the operator locality $l \leq k_{op}$ is given by the number of ways to pick $l$ out of $L$ sites times the possible configurations of the Pauli matrices on each of these non-trivial sites:

$$N_{loc} = \sum_{l=1}^{k_{op}} 3^l \binom{L}{l} \approx 3^{k_{op}} \frac{L^{k_{op}}}{(k_{op})!} = 3^{k_{op}} \frac{\left(\log_2 D\right)^{k_{op}}}{(k_{op})!}. \tag{56}$$

Note that the number of local operators in the second definition, $(k_{op}, k_{sp}) = (k_{op}, k_{op})$, is strictly lower than this estimate.

In the following, we focus on two examples of $s = 1/2$ spin chains: The mixed field Ising spin chain, which has chaotic and integrable regions in its parameter space, and the XYZ Heisenberg spin chain, which is integrable. We start by describing the Hamiltonians and their symmetries in section 4.1.1 and 4.1.2 and turn to the numerical analysis of their complexities in section 4.1.3.

### 4.1.1 Ising spin chain

The mixed field Ising model is described by the Hamiltonian

$$H_{\text{Ising}} = -\sum_j [S_z^{(j)} S_z^{(j+1)} + h_x S_x^{(j)} + h_z S_z^{(j)}], \tag{57}$$

where $h_x$ and $h_z$ are the magnetic fields in the transverse and longitudinal direction respectively. The site $L + 1$ is periodically identified with the first site, $S_a^{(L+1)} \equiv S_a^{(1)}$. The system is trivially integrable when $h_x = 0$, since the Hamiltonian is constructed out of mutually commuting terms. The eigenvalues of this Hamiltonian can therefore be readily read off as a function of $h_z$. More interestingly, the model is also integrable along the line $h_z = 0$. This fact is less straightforward to see, but very well-known from the dual picture as a theory of free fermions which can be obtained after applying a Jordan-Wigner transform [44, 45] (see [46] for a review).

Nielsen's complexity of a trivially integrable Hamiltonian such as the longitudinal Ising model is very non-generic and, in fact, almost as trivially solved. A first hint for the peculiar behavior of these models is the proliferation of exact null directions of the $Q$-matrix as the locality threshold increases. This can be manifested as follows: since the Hamiltonian (57) is diagonal in the same basis as any generator (55) made out of Pauli-$z$ operators, the diagonal of each of these operators defines an eigenvector of the $Q$-matrix with eigenvalue 0 or 1, following (30) and (31). The details depend on the locality that is chosen. All these generators, and hence in particular the local generators, are very simple diagonal matrices. Their eigenvalues are, up to an overall normalization factor, equal to 1 or $-1$. Such a structure allows some of the integer vectors $\mathbf{k}$ in (26) to point in null directions of the $Q$-matrix, hence contributing very little to the complexity. This provides a way to reduce the coefficients $E_n t$ by combinations of integer subtractions (up to factors of $2\pi$) almost as efficiently as in the bi-invariant metric (18). Combining many subtractions in purely local directions, when possible, will generally be more effective than forcing the $E'_n$ to lie exactly between $-\pi$ and $\pi$ using (19), since this latter approach generally induces large penalties due to contributions from nonlocal directions. The resulting complexity curve for the longitudinal Ising model is independent of the cost factor $\mu$. This situation is very similar to the integrable SYK models considered in [5, 6], where this approach to solving for Nielsen's complexity is explained in more detail.

To bring in a more interesting setting, we shall mostly focus on the transverse Ising model as a representative for integrable systems in our subsequent numerics. The Ising Hamiltonian (57) is known to be chaotic away from the integrable line $h_z = 0$, and to exhibit strongly chaotic behavior around the point $(h_x, h_z) = (-1.05, 0.5)$ [47, 48].

The periodic Ising Hamiltonian displays a large number of degeneracies due to globally conserved charges. To facilitate the computation of the enlarged $\tilde{Q}$-matrix introduced in section 2.2 to deal with degeneracies, we shall often diagonalize the Hamiltonian simultaneously with at least one of these conservation laws to split some of these degeneracies prior to computing the $\tilde{Q}$-matrix. Let us therefore recall the global conservation laws of (57) for future reference. First, the Hamiltonian is translation invariant. The generator of this symmetry is the momentum operator

$$P = i \log T \,, \tag{58}$$

with $T$ the translation operator

$$T = \mathcal{P}_{L,L-1} \mathcal{P}_{L,L-2} \dots \mathcal{P}_{L,1} \,, \tag{59}$$

constructed from the permutation operator $\mathcal{P}_{i,j}$ which transforms any Pauli operator acting on site $i$ to the same Pauli operator acting on site $j$. For spin $1/2$, the permutation operator takes the form

$$\mathcal{P}_{i,j} = \frac{\sum_a S_a^{(i)} S_a^{(j)} + \mathbf{I}}{2} \,, \tag{60}$$

where $a$ runs over the three indices of the Pauli matrices. Second, the Hamiltonian commutes with the parity operator

$$\Pi = \mathcal{P}_{1,L} \mathcal{P}_{2,L-1} \dots \mathcal{P}_{\lfloor L/2 \rfloor, \lceil L/2+1 \rceil} \,, \tag{61}$$

which represents the symmetry of (57) under reflections across the midpoint of the chain. By conjugating (61) with the translation operator (59), the Hamiltonian is shown to be symmetric under a reflection about any chosen point. When $h_z = 0$, the spin flip operator

$$S = \prod_i S_x^{(i)} \,, \tag{62}$$

becomes an additional symmetry of the Hamiltonian.

Integrable spin chains have been studied in great detail. From all the attractive features they possess, we shall be most interested in the towers of conserved charges and, in particular, the role they play in reducing the complexity of the Hamiltonian evolution. In addition to the global conservation laws listed above, the transverse Ising model possesses two towers of 'local' conserved charges[15] in involution [50] that render the model integrable:

$$I_{2l-1} = \sum_j \left( S_y^{(j)} S_x^{(j+1)} S_x^{(j+2)} \ldots S_x^{(j+l-1)} S_z^{(j+l)} - S_z^{(j)} S_x^{(j+1)} S_x^{(j+2)} \ldots S_x^{(j+l-1)} S_y^{(j+l)} \right), \qquad (63)$$

$$I_{2l} = \sum_j \left( S_z^{(j)} S_x^{(j+1)} S_x^{(j+2)} \ldots S_x^{(j+l)} S_z^{(j+l+1)} - h_x S_y^{(j)} S_x^{(j+1)} S_x^{(j+2)} \ldots S_x^{(j+l-1)} S_y^{(j+l)} \right.$$
$$\left. - h_x S_z^{(j)} S_x^{(j+1)} S_x^{(j+2)} \ldots S_x^{(j+l-1)} S_z^{(j+l)} + S_y^{(j)} S_x^{(j+1)} S_x^{(j+2)} \ldots S_x^{(j+l-2)} S_y^{(j+l-1)} \right), \qquad (64)$$

for $l > 1$ and

$$I_1 = \sum_j \left( S_y^{(j)} S_z^{(j+1)} - S_z^{(j)} S_y^{(j+1)} \right), \qquad (65)$$

$$I_2 = \sum_j \left( S_z^j S_x^{(j+1)} S_z^{(j+2)} - h_x S_y^{(j)} S_y^{(j+1)} - h_x S_z^{(j)} S_z^{(j+1)} - S_x^{(j)} \right). \qquad (66)$$

A relevant aspect of these conserved charges, for our purposes, is their locality degree in terms of $k_{op}$ and $k_{sp}$. As explained in section 2.1, at a certain locality threshold $(k_{sp}^{max}, k_{sp}^{max})$ each exact null eigenvalue of the $Q$-matrix indicates a conservation law that has a locality degree $k_{op} \leq k_{op}^{max}$ and $k_{sp} \leq k_{sp}^{max}$. The individual terms in the conservation laws (63)-(66) act on spatially dense regions, i.e. there are no gaps between sites where they act nontrivially. Specifically, for the odd tower one has $k_{sp}(I_{2l-1}) = k_{op}(I_{2l-1}) = l+1$ and for the even tower one finds $k_{sp}(I_{2l}) = k_{op}(I_{2l}) = l+2$. We therefore expect at least[16] two additional zero eigenvalues of the $Q$-matrix to appear whenever $k_{op}$ is increased by one unit, independently of how $k_{sp}$ is varied (as long as it remains larger than $k_{op}$ at every step). Note that the operators (63)-(66) respect the translation invariance and are symmetric under the action of the spin flip operator (62), but do not commute with the reflection operator (61).

### 4.1.2 XYZ Heisenberg spin chain

As a second illustration, we shall consider the XYZ Heisenberg spin chain which is described by the Hamiltonian

$$H_{XYZ} = \sum_j \left[ J_x S_x^{(j)} S_x^{(j+1)} + J_y S_y^{(j)} S_y^{(j+1)} + J_z S_z^{(j)} S_z^{(j+1)} \right], \qquad (67)$$

with constants $J_x$, $J_y$ and $J_z$. When $J_x = J_y$, the system is known as the XXZ chain, while the isotropic limit $J_x = J_y = J_z$ is referred to as the XXX chain. Both models have an explicit $U(1)$ symmetry of rotations in the $xy$-plane generated by the angular momentum in the $z$-direction

$$\mathcal{J}^z = \sum_{j=1}^{L} S_z^{(j)}. \qquad (68)$$

This symmetry group gets enhanced to $SU(2)$ when all the coefficients are equal. As usual in quantum mechanics with global symmetries, the spectrum splits into sectors defined by the

---

[15]We thank Jacques Perk for drawing our attention to the connection between these conservation laws and the Onsager algebra [49].

[16]An exact counting requires considering also linearly independent products of conservation laws of lower locality.

various charges. In the presence of 3-dimensional rotation symmetry, the spectrum of the XXX model splits into sectors of fixed (68) and total angular momentum

$$\mathcal{J}^2 = (\mathcal{J}^x)^2 + (\mathcal{J}^y)^2 + (\mathcal{J}^z)^2 \,, \tag{69}$$

where $\mathcal{J}^x$ and $\mathcal{J}^y$ are defined in analogy with (68). Similarly to the Ising model, the Heisenberg Hamiltonian (67) also enjoys momentum (58) and parity (61) conservation for any choice of coefficients $J_i$.

All of the Heisenberg Hamiltonians (67) are integrable. An explicit tower of conserved charges is known as a function of the coefficients $J_i$ [51]. Here again, the locality properties of the conservation laws are their most relevant features. As an illustration, the conserved charge of lowest locality degree, beyond the Hamiltonian, is given by

$$I_1 = \sum_j \left( \hat{\mathbf{s}}^{(j)} \times \tilde{\mathbf{s}}^{(j+1)} \right) \cdot \hat{\mathbf{s}}^{(j+2)} \,, \tag{70}$$

with

$$\hat{S}_a^{(j)} \equiv \sqrt{J_a} S_a^{(j)} \,, \qquad \tilde{S}_a^{(j)} = \sqrt{\frac{J_x J_y J_z}{J_a}} S_a^{(j)} \,. \tag{71}$$

The boldface letter is a short-hand notation for the spatial vector with components $\mathbf{S}^{(j)} = (S_x^{(j)}, S_y^{(j)}, S_z^{(j)})$. The locality of (70) can be read off directly as $k_{op} = k_{sp} = 3$. The operator (70) is the first in a tower of conserved charges for the XYZ Hamiltonian (67), analogous to (63) or (64). The expression for a generic element in this tower can be found recursively and we refer the reader to [51] for their specific form. (An explicit construction of these charges has more recently been put forward in [52].) As for (63) or (64), it can be shown that the term of largest locality in every member of this tower has $k_{sp} = k_{op}$, and the locality degree increases by one unit along the tower. In contrast to the transverse Ising model, the integrable structure of the Heisenberg model is defined by a single tower of conservation laws.

Finally, we shall consider a chaotic representative obtained by breaking the integrability of the Hamiltonian (67) through the addition of a magnetic field in the $z$-direction,

$$H_{XYZ}^{ch} = \sum_j [J_x S_x^{(j)} S_x^{(j+1)} + J_y S_y^{(j)} S_y^{(j+1)} + J_z S_z^{(j)} S_z^{(j+1)} - h_z S_z^{(j)}] \,. \tag{72}$$

As described e.g. in [47] for the Ising spin chain, the transition to a fully chaotic regime happens gradually as one moves away from the 3-dimensional integrable parameter subspace at $h_z = 0$. We will ensure where necessary that the Hamiltonian is well within the chaotic regime, as diagnosed by means of e.g. a spectral statistics analysis.

### 4.1.3 Complexity reduction and integrability at $s = 1/2$

We are now ready to present the results of our numerical analysis for the behavior of our variational Ansatz (26) for complexity in chaotic and integrable spin 1/2 chains. The complexity curves we shall display are obtained by approximately solving (26) at every time step using the set of numerical methods described at the end of section 2.1. The evolution of this probe of complexity has a universal early time linear growth, whose slope was fixed to 1 in (20)-(21). Some information about the integrability of a model can be reflected in occasional sharp reductions along the linear growth, as was observed in [6]. In this work, however, we concentrate on the complexity reduction of integrable models visible in the late-time saturation region.

The first step in the numerical scheme to solve (26) is to compute the $Q$-matrix (27) for the considered Hamiltonian, given a choice of locality threshold to define the local generators. As discussed in section 2, there is an ambiguity along the way to (26) when applying the

variational method on periodic spin chains due to the degeneracies in the energy spectrum. We shall fix this ambiguity by means of the procedure outlined in section 2.2. The MATLAB implementation can be found in [43]. To reduce the resources needed for this computation, we systematically split some of the degeneracies prior to computing the enlarged $\tilde{Q}$-matrix by simultaneously diagonalizing the Hamiltonian with the momentum operator (58). When applicable, we shall also divide the eigenvectors in sectors of definite angular momentum (68) and (69). On general grounds, we expect that the tower of conservation laws of the corresponding integrable models will respect all these symmetries. We note, however, that a residual ambiguity can remain following the eigenbasis-fixing procedure of section 2.2, although we expect the variations in complexity to be small for different eigenbases, provided they all produce $Q$-matrices with the largest possible kernel at fixed locality threshold. Once the $Q$-matrix is obtained in the 'optimal' energy eigenbasis, we apply the LLL algorithm to improve the orthogonality properties of the standard lattice basis with respect to the inner product specified by the $Q$-matrix. This step only needs to be done once and the output can be used for all times. Afterwards, we apply the Babai and greedy algorithms, as discussed in detail in [6] and reviewed in section 2.1, using the LLL-reduced basis in order to determine which lattice point the vector $\mathbf{E}t$ is closest to. This computation needs to be performed at every time step, for times that we verified to be well within the saturation region of the complexity probe. Specifically, we will sample time points between $t = 5 \times 10^7$ and $t = 6 \times 10^7$ in steps of $\Delta t = 10^4$. From each sample, we shall determine the mean saturation height $\mathcal{C}^{\text{sat}}$ and this will constitute our characterization of the late-time complexity for the evolution operator of a dynamical system. The Hilbert space dimension and cost factor will be set to $\mu = D = 2^{12}$.

Before delving into the detailed analysis, we summarize the different perspectives that we will concentrate on to verify the connection between the zero (and small) eigenvalues of the $Q$-matrix and the complexity reduction in integrable models. We start with studying the correlation between a suitable choice for the set of local operators, which can e.g. be aligned with the patterns in the conservation laws in integrable systems, and complexity reduction using the Ising and Heisenberg XYZ Hamiltonians. The nearest-neighbor character of the spin chain interactions under consideration allows for several natural ways to separate easy and hard generators as a function of a locality parameter $k$. For this, we shall exploit the freedom in varying $k_{op}$ independently of $k_{sp}$ in specifying the locality threshold. The different definitions we shall focus on share an identical increase in the number of local conservation laws as $k$ grows. From our working hypothesis, which connects complexity reduction to the size of the kernel of the $Q$-matrix, we expect the complexity reduction to be pretty much insensitive to different definitions, at fixed $k$, even when the size of the set of easy generator increases. (As we argued in section 3, increasing the number of local operators decreases the height of the complexity plateau for generic systems. The complexity reduction due to integrable structures, however, dominates over this effect.) In contrast, we expect that relabeling a small subset of easy generators, chosen in such a way that one of the local conservation laws no longer remains local, as hard generators will have a direct impact on the height of the late-time complexity plateau in integrable models. Besides the complexity curves, we shall also be interested in observing the influence of these modifications on the shape of the $Q$-eigenvalue distribution. Additionally, it is interesting to ask whether models with a larger global symmetry group experience a stronger complexity reduction. Hence, we subsequently turn to the XXZ Heisenberg Hamiltonian and compare results with XYZ. Their structures of conservation laws differ by a $U(1)$ rotation symmetry, which adds one local conservation law. We therefore expect this to lead to a larger complexity suppression. To conclude, we provide evidence for the qualitative validity of RMT results (44) and (49) derived in the previous section, which predict the first moments of physical chaotic $Q$-eigenvalue distributions. We also verify that the estimate (53) can be used to predict the saturation height of the upper bound on complexity

of chaotic models, and investigate how well it performs for integrable models.

#### 4.1.3.1 Complexity reduction in integrable Ising and Heisenberg spin chains

One of the main insights gained from [6] is that the difference in the upper bound on complexity between integrable and chaotic models becomes more pronounced as the locality threshold $k$ is increased. Although it is conventional to choose at most two-body operators to be local in Nielsen's complexity, it was found to be instructive to relax this condition. As we discussed in section 3, the late-time saturation height of the complexity is generally expected to decrease regardless of the dynamics when increasing the locality threshold, simply because the number of local operator increases. However, on top of the reduction resulting from these general arguments, one heuristically expects that the complexity of integrable evolution goes down much faster at larger locality thresholds. We expect this to happen by means of shortcuts induced in the minimization problem (26) by the presence of integrable conserved charges (arranged in towers of increasing locality degree) and the changes in the $Q$-matrix they bring along. This motivates explorations of complexity saturation for various choices of locality thresholds.

*Sets of local operators $\mathcal{T}_1(k)$ and $\mathcal{T}_2(k)$*

As discussed at the beginning of this section, there are several ways to increase the locality threshold and hence the size of the set of local operators. We shall adopt three distinct definitions and examine their effect on the complexity. First, we consider $k_{op} = k_{sp}$ and vary both parameters simultaneously. We shall denote as $\mathcal{T}_1(k)$ the set of local operators according to the threshold $k_{op} = k_{sp} = k$. In light of the structure of the tower of conservation laws (63)-(64), this choice appears as the most economical for producing large numbers of local conservation laws in integrable spin chains. Indeed, as $k_{op}$ increases by 1 unit, the kernel of the $Q$-matrix likewise increases by 1 or 2 units in the Heisenberg and Ising model, respectively. As we explained above, we want to examine in detail whether a larger $Q$-matrix kernel is a proxy for lower complexity. If this interpretation is correct, allowing additional generators to be local without gaining new conservation laws should not significantly lower the complexity. Conversely, removing a few generators from $\mathcal{T}_1(k)$ may upset the characterization of some conservation laws as local. Assuming that the saturation height is mainly influenced by additional exact null directions in the $Q$-matrix, this would be expected to result in an increased complexity.

To test the above ideas, we define a second notion of locality for separating easy and hard directions whereby $k_{sp}$ is fixed to some intermediate value and we gradually increase $k_{op}$ up to $k_{sp}$. In particular, we shall use $k_{sp} = 6$. Beyond $k_{sp} = 7$, both chaotic and integrable spin chains at $L = 12$ would develop an additional exact zero $Q$-eigenvalue for $k_{op} > 3$, due to $H^2$ becoming local. Since we are interested in precisely understanding the influence of the kernel of the $Q$-matrix on complexity reduction, we fix $k_{sp}$ to a smaller value for simplicity. We denote with $\mathcal{T}_2(k)$ the set of local generators at locality threshold $(k_{op}, k_{sp}) = (k, 6)$. Note that at a given $k \leq 6$, the local generators in $\mathcal{T}_1(k)$ are all contained in $\mathcal{T}_2(k)$. We shall examine whether this larger set of local operators causes a further complexity reduction.

*Numerical results for the complexity bound for the integrable Ising Hamiltonian*

As mentioned above, we shall mainly be interested in the saturation values of the complexity at late times. For completeness, however, we illustrate the typical shape of a complexity curve in Figure 1. The main features are an early time growth with unit slope that goes on until saturation. The plateau height is visibly lower for the integrable instance, and the fluctuations around the mean are larger. One can see from the right plot that the chosen time window

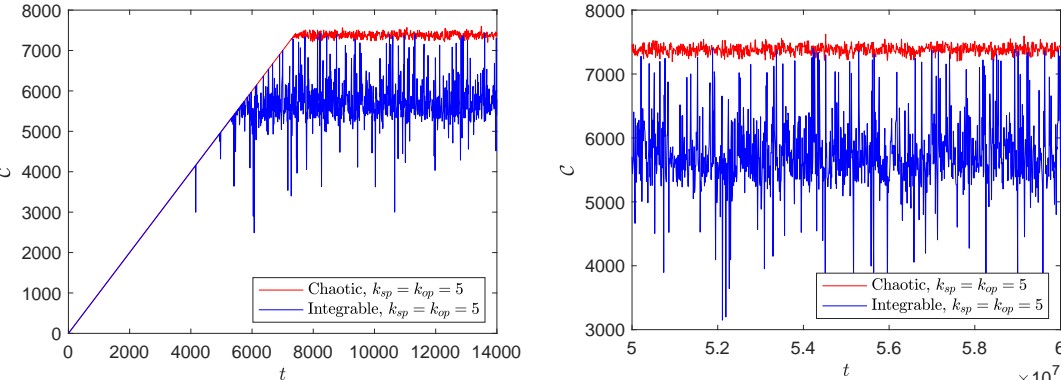

Figure 1: **Left:** The time evolution of the upper bound on Nielsen's complexity for the Hamiltonian evolution of the transverse Ising model at $(h_x, h_z) = (-1.05, 0)$ (blue) and the Ising model evaluated at a chaotic parameter point $(h_x, h_z) = (-1.05, 0.5)$ (red). The length of the chain is $L = 12$. To avoid using candidate minima that are evidently sub-optimal in estimating the complexity (26), at every time step we compare the output of the minimization procedure to the complexity associated to the early time solution $k_n = 0$ and the minimum of the two is chosen. **Right:** A zoom on the time window inside the plateau region used in the numerics.

used in our subsequent numerical analysis seems representative for the saturation value of the complexity probe.

Given the important role of the $Q$-matrix in estimating the saturation height of the complexity curve, we start our discussion by displaying the $Q$-eigenvalue distributions for the integrable and chaotic representatives of the Ising model as a function of $k_{op} = k_{sp}$, as well as for $k_{sp} = 6$ as a function of $k_{op}$ in Figure 2. The size of the $Q$-matrix kernel is highlighted in the left corner and one can notice that the number of null directions grows as expected from (63)-(66) for the integrable model. The chaotic model has a single zero $Q$-eigenvalue for all locality thresholds, corresponding to the Hamiltonian itself.[17] Furthermore, the initially (at $k_{op} = 2$) sharply peaked $Q$-eigenvalue distribution of the transverse Ising model can be seen to flatten towards lower values as $k_{op}$ increases. This implies that many directions in the manifold of unitaries have a diminished cost compared to the lattice distance measure associated to the chaotic distributions. This is a first indication of the ensuing complexity reduction for the integrable representative. Comparing the upper and lower plots, one observes a slightly larger spread towards small eigenvalues for $\mathcal{T}_2(k)$ compared to $\mathcal{T}_1(k)$. This is not surprising since the size $N_{loc}$ of the former set is larger (except at $k_{op} = k_{sp} = 6$, where the two sets are identical). Indeed, a shifted distribution is visible in the chaotic and integrable histograms. To investigate this in more detail, we display the growth of $N_{loc}$ with the locality threshold $k$ for $\mathcal{T}_1(k)$ and $\mathcal{T}_2(k)$ in Figure 3. From the RMT arguments of section 3, the mean of the $Q$-eigenvalue distributions of chaotic Hamiltonian eigenbases are expected to be essentially following a qualitative behavior set by $N_{loc}(k)$. This expectation is verified when comparing the left and middle plot in Figure 3. The evolution of the mean of the $Q$-eigenvalue distribution for the transverse Ising model nevertheless shares the same qualitative features as the chaotic model. The main difference between integrable and chaotic is the range of the vertical axis in the middle and right plot of Figure 3.

We now turn to the saturation values of the complexity curves. The plateau heights for the upper bound on Nielsen's complexity associated to these $Q$-eigenvalue distributions are

---

[17]Note that the identity operator was chosen to be nonlocal, so that the associated trivial null direction (corresponding to the all-one vector) would not interfere with our analysis.

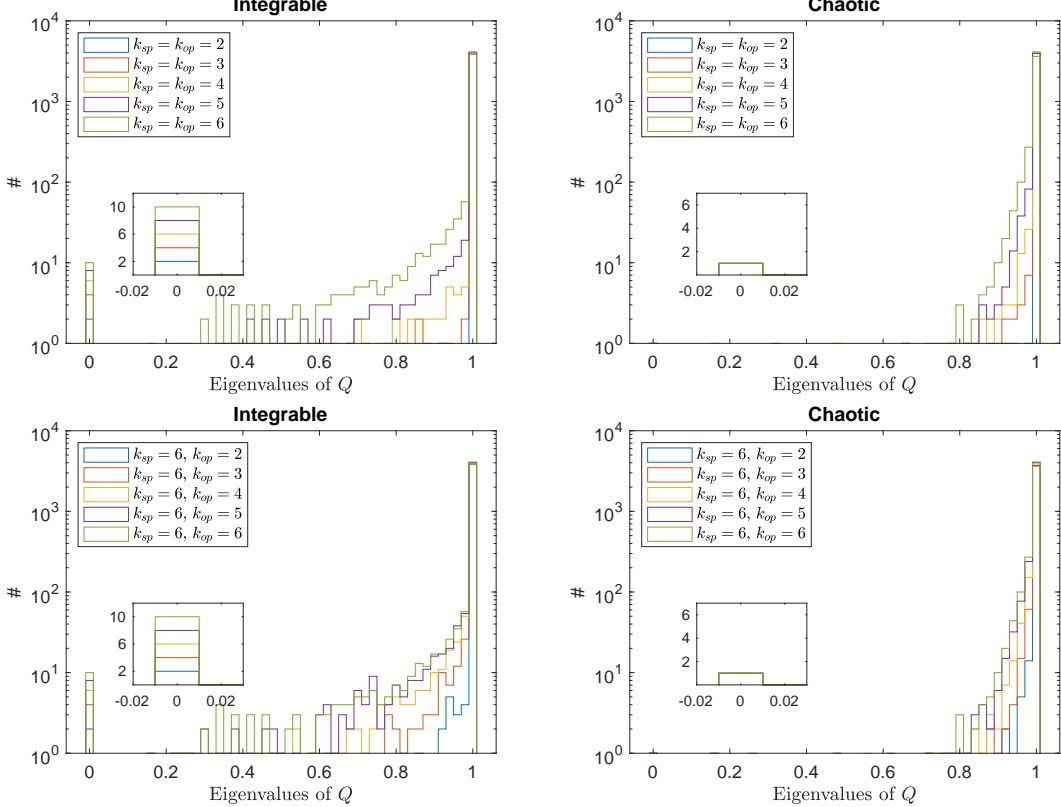

Figure 2: Histograms of the eigenvalues of the Q-matrix for **Left column:** an integrable Ising spin chain (57) with $(h_x, h_z) = (-1.05, 0)$ and **Right column:** a chaotic Ising spin chain with $(h_x, h_z) = (-1.05, 0.5)$ for varying locality thresholds as **Top row:** $k_{sp} = k_{op}$ or **Bottom row:** only varying $k_{op}$ while keeping $k_{sp} = 6$ fixed. The length of the chain is set to $L = 12$. In the bottom left inset of each plot we zoom in on the size of the kernel of the $Q$-matrix. It can be seen that in the integrable case, the number of zero eigenvalues increases in steps of 2, in accordance with the form of the conserved charges (63)-(66).

numerically estimated by averaging the approximate solutions to (26) sampled over a late time-interval. The comparison between both notions of locality is displayed in Figure 4. First, one observes a substantial complexity reduction for the transverse Ising model as $k$ increases, which is already visible at $k = 2$ due to the additional zero eigenvalue of the $Q$-matrix compared to the chaotic model. In addition, the curves for the two types of locality defined by $\mathcal{T}_1$ and $\mathcal{T}_2$ are found to be almost indistinguishable, which provides strong evidence for the dominant role of the local conservation laws and associated zero $Q$-eigenvalues in the complexity reduction mechanism. In particular, despite the fact that $N_{loc}$ greatly increases in going from $\mathcal{T}_1$ to $\mathcal{T}_2$, as can be seen from the first figure in Figure 3, this change only decreases the complexity by a very small amount (which appears as important as the decrease in the chaotic case) compared to the large decrease already observed when declaring the operators in $\mathcal{T}_1$ local.

*Numerical results for the complexity bound for the XYZ Heisenberg Hamiltonians*

As a second example, we repeat this analysis for integrable and chaotic Heisenberg chains. We find that the Hamiltonian (72) with parameters $(J_x, J_y, J_z, h_z) = (-0.35, 0.5, -0.1, 0.8)$ is well within the chaotic regime and consider the integrable XYZ Heisenberg Hamiltonian with

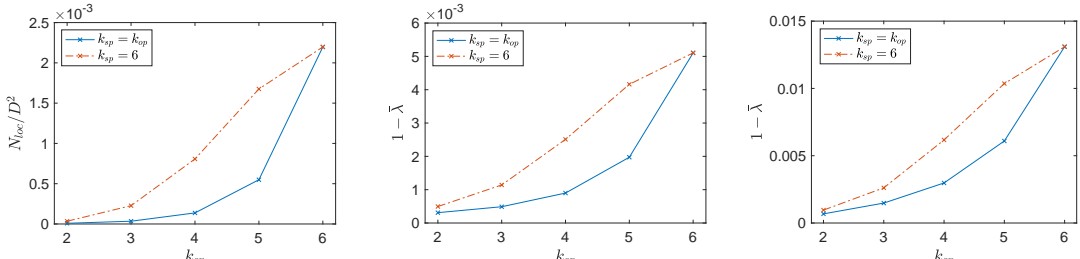

Figure 3: **Left:** The fraction of local generators to the total number of generators, as a function of $k_{op}$ for the two types of locality thresholds $\mathcal{T}_1$ and $\mathcal{T}_2$ we impose in the main analysis. **Middle and Right:** The distance between the mean of the $Q$-eigenvalue distribution and the right edge of the $Q$-histograms, for, respectively, the chaotic and integrable Ising model as a function of $k_{op}$. The shape of the curves in all three plots is very similar. The results for the chaotic model qualitatively agree with the RMT prediction (44) up to a factor of order one.

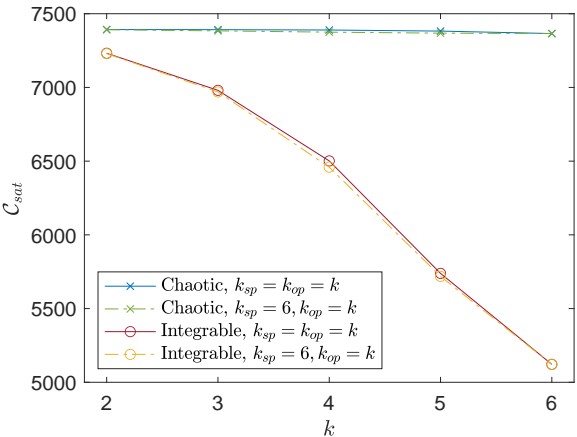

Figure 4: The late-time saturation value of the complexity bound for the integrable $(h_x, h_z) = (-1.05, 0)$ and chaotic $(h_x, h_z) = (-1.05, 0.5)$ Ising model with $L = 12$ sites as a function of a locality threshold specified by $k$. The dashed line corresponds to $k_{sp} = 6$ fixed and varying $k_{op}$, while for the solid line we vary both locality degrees $k_{sp} = k_{op}$.

the same coefficients $J_i$. The corresponding $Q$-eigenvalue distributions and complexity saturation values as a function of the locality threshold are shown in Figures 5 and 6, respectively. These results very much support the conclusions we reached by means of the Ising spin chain. The main difference between the Ising and Heisenberg chains is the magnitude of the complexity decrease for the integrable instances. A close inspection of the range of the vertical axis in Figure 4 and Figure 6 reveals an evident correlation between complexity reduction and integrable structures. The transverse Ising model indeed possesses a second tower of conservation laws compared to the Heisenberg model, and this causes the height of the complexity plateau of the Ising chain to decrease visibly faster with $k_{op}$ than the Heisenberg chain.

The smaller range of the complexity in the Heisenberg model allows us to analyze the difference in complexity between the two locality definitions in more detail and ask whether the suppression observed when enlarging $\mathcal{T}_1$ to $\mathcal{T}_2$ has the same origin in the integrable and chaotic systems. In Figure 6, one can notice that the difference between the two chaotic curves, as a function of $k_{op}$, is very similar to the difference between the two integrable curves and

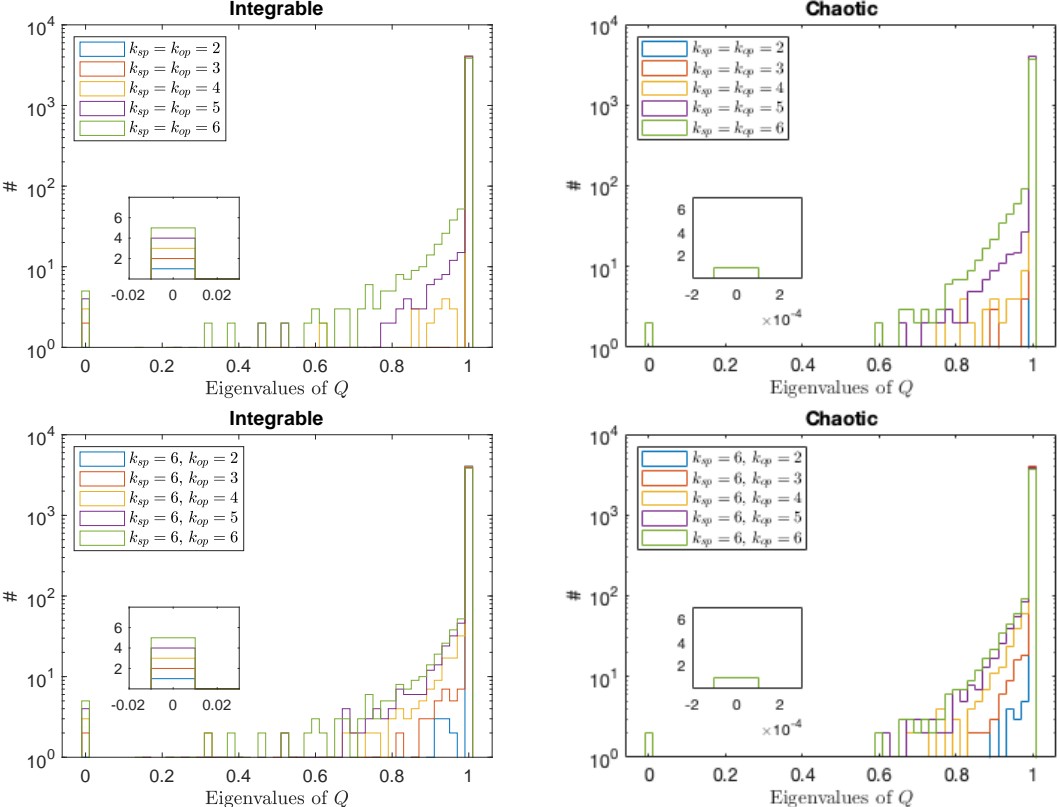

Figure 5: Histograms of the eigenvalues of the Q-matrix for the $L = 12$ **Left column:** integrable XYZ model (67) and for the **Right column:** chaotic XYZ model with magnetic field (72) (right) for varying locality thresholds as **Top row:** $k_{sp} = k_{op}$ or **Bottom row:** only varying $k_{op}$ while keeping $k_{sp} = 6$ fixed. For the coupling constants we used the numbers $(J_x, J_y, J_z) = (-0.35, 0.5, -0.1)$ and on the right plot we chose $h_z = 0.8$. In the bottom left corner of each plot we zoom in on the zero eigenvalues. The number of zero $Q$-eigenvalues in the integrable case increases in accordance with the form of the conserved charges.

hence of the same origin. This suggests that the additional reduction in complexity when going from $\mathcal{T}_1$ to $\mathcal{T}_2$ observed in the integrable chain can indeed be solely attributed to enlarging the set of local operators, with no apparent connection to the specifics of the integrable structures.

*A third set of local operators $\mathcal{T}_3(k)$*

The previous discussion provides compelling evidence that the complexity plateau height is insensitive to the addition of elements to the set of easy generators that do not result in new local conservation laws. To tackle our working hypothesis from another angle, we consider the locality threshold $k_{op} = k_{sp}$ and do the following. Instead of enlarging $\mathcal{T}_1$ by adding generators that are not producing any new local conservation law, we pursue the opposite strategy. We consider removing a very small number of generators from $\mathcal{T}_1$ and observe the effect on the complexity and $Q$-matrices. Specifically, we define a third set of local generators $\mathcal{T}_3(k)$ defined by removing from $\mathcal{T}_1(k)$ a subset of operators of size $3^k$, containing generators of the form

$$S_{i_1}^{(l)} S_{i_2}^{(l+1)} \cdots S_{i_k}^{(l+k-1)}, \tag{73}$$

for a specific choice of $l$, and where the Pauli operators are understood to act non-trivially on each site. Note that $3^k$ is small compared to $N_{loc}(k)$. The ratio in fact decreases with

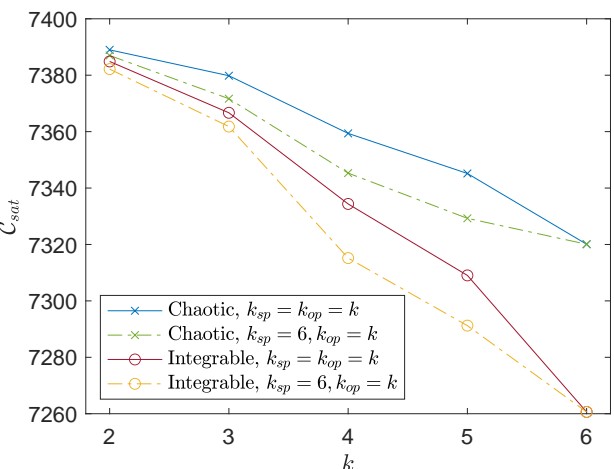

Figure 6: The late-time saturation value of the complexity bound for the integrable XYZ model (67) and the chaotic Hamiltonian with magnetic field (72) for $L = 12$. For the coupling constants we used the numbers $(J_x, J_y, J_z) = (-0.35, 0.5, -0.1)$ in both cases and $h_z = 0.8$ for the chaotic Hamiltonian.

increasing $k$. The removal of (73) from the set of local generators $\mathcal{T}_1(k)$ directly affects the kernel of the $Q$-matrix, since these generators belong to the set of operators of largest locality degree appearing in the conservation laws of the integrable systems. For chaotic models, we do not expect this modification to impact visibly any of the results displayed above.

In Figure 7, we display the $Q$-matrix eigenvalue distribution for the chaotic and integrable Ising model for the locality specification corresponding to $\mathcal{T}_3(k)$. The histograms can in fact be seen to experience few changes compared with the upper plots in Figure 2. Although the number of exact null direction has decreased accordingly, the number of small eigenvalues nevertheless remains largely unchanged, except perhaps for the purple curve at $k_{op} = 5$. It is natural to suspect that the two eigenvalues that were previously exact zeros have been incorporated in the intermediate $Q$-eigenvalues after the removal of (73), without changing the overall distribution too much. This conclusion is supported by Figure 8 on the right, where the mean of the $Q$-eigenvalue distributions are computed as a function of $k$. The curve for $\mathcal{T}_1(k)$ is almost identical to $\mathcal{T}_3(k)$. We therefore conclude that the $Q$-eigenvalue distribution is not very sensitive to the removal of (73) from $\mathcal{T}_1(k)$. In contrast, the modified locality does lead to an substantial increase in complexity for the integrable Ising model, as shown in the left plot of Figure 8. This is interesting, since the change in the complexity has not been signaled by the distribution of eigenvalues of the $Q$-matrix. This observation suggests that the kernel of the $Q$-matrix has a fundamental role in the complexity reduction.

### 4.1.3.2 Complexity of XYZ vs XXZ Heisenberg models

The integrability of the Heisenberg model for any choice of coefficients $J_i$ offers a useful arena to test various hypotheses. On top of the tower of conservation laws that is common to all the variants of the Heisenberg model, the XXX and XXZ chains enjoy additional global symmetries which do not originate from the integrable structures of the model. It is therefore interesting to investigate whether these additional symmetries influence the complexity curve in any way. Since the $SU(2)$ symmetry of the XXX chains induces inconveniently large degenerate energy subspaces, we shall restrict our attention to XXZ for simplicity.

Let us first reflect on what we expect to find. The XXZ spin chain has an additional $U(1)$ symmetry of rotations with generator $\mathcal{J}^z$ (68), which is a local conserved operator.

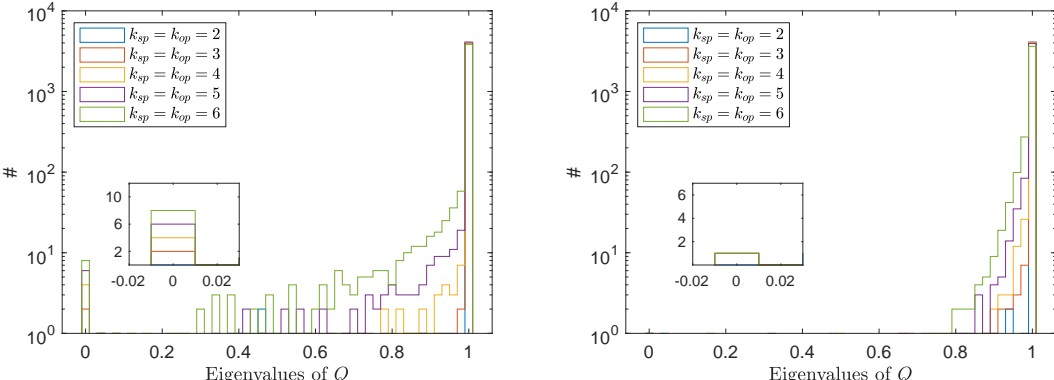

Figure 7: Histograms of the $Q$-eigenvalue distributions as a function of $k$ for a set of local generators $\mathcal{T}_3(k)$. **Left:** The integrable transverse Ising model at $(h_x, h_z) = (-1.05, 0)$. **Right:** A chaotic representative of the Ising model at $(h_x, h_z) = (-1.05, 0.5)$. In both cases, the length of the chain is set to $L = 12$.

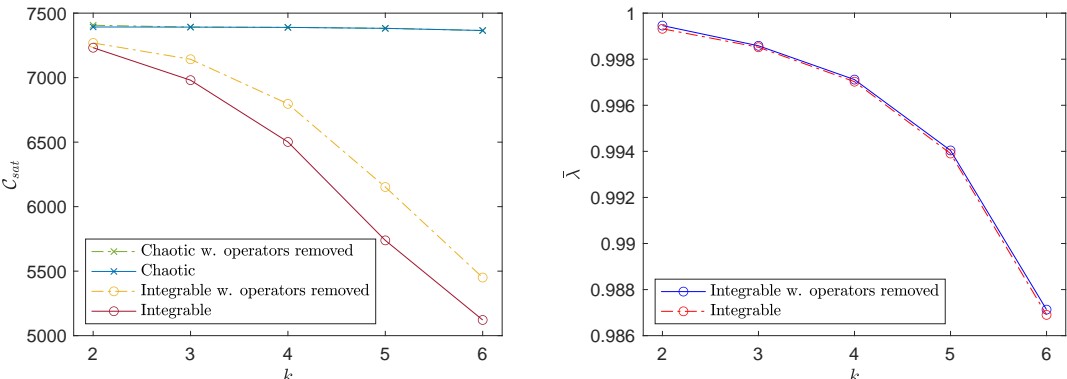

Figure 8: **Left:** Saturation values of the complexity bound as a function of $k$ for a set of local generators $\mathcal{T}_1$, with and without the operators (73). This is displayed for chaotic and integrable instances of the Ising model at $L = 12$. **Right:** The mean of the $Q$-eigenvalue distributions shown in the left plot of Figure 7 and the upper left plot of Figure 2.

One can therefore anticipate a systematically lower complexity plateau height for XXZ compared to XYZ.

We find this to be well verified by the numerics, as shown in Figure 9. At $k = 2$, the global symmetry $\mathcal{J}^z$ (68) joins the Hamiltonian in the set of local operators for the XXZ chain. As anticipated, this results in a slightly lowered complexity compared to the XYZ chain, which has the Hamiltonian as a single local conservation law. As $k$ increases, the number of local conserved operators gradually increases in both systems and complexity plateau heights for XYZ at $k$ seems to match the complexity plateau heights for XYZ at $k + 1$. At $k = 6$, the XXZ chain enjoys one more addition to the kernel of the $Q$-matrix compared to the XYZ chain. This comes from the possibility to construct linearly independent conservation laws of higher locality by means of products of conserved operators of lower locality, in particular because the locality of $\mathcal{J}^z$ is very low, $k_{op} = k_{sp} = 1$. This can be observed to directly affect the complexity, which drops slightly more for the XXZ chain in going from $k = 5$ to $k = 6$.

We can therefore conclude that all our findings have consolidated the hypothesis connecting the complexity reduction in integrable models to the size of the kernel of the $Q$-matrix.

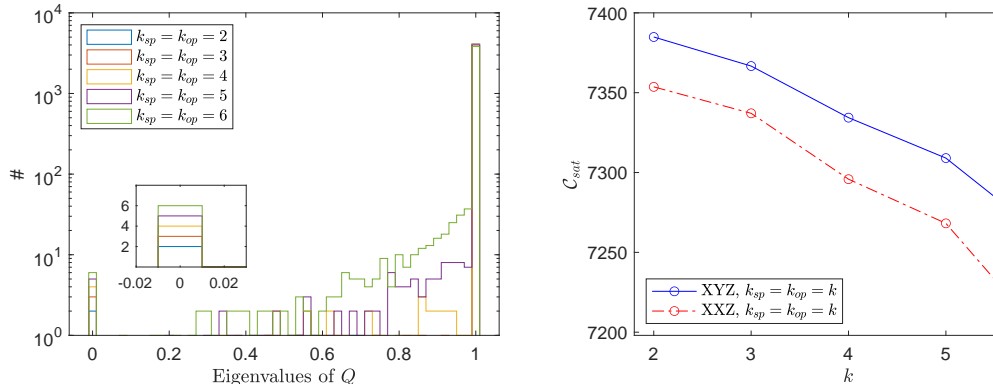

Figure 9: **Left:** The distribution of eigenvalues of the Q-matrix for the integrable XXZ model at $(J_x = J_y, J_z) = (-0.35, -0.1)$ for increasing $\mathcal{T}_1(k)$, for $L = 12$. In the bottom left corner, we zoom in on the number of exact zero eigenvalues. **Right:** A comparison between the late-time saturation values of the complexity bound as a function of $\mathcal{T}_1(k)$ for the integrable XYZ at $(J_x, J_y, J_z) = (-0.35, 0.5, -0.1)$ and XXZ model at $(J_x = J_y, J_z) = (-0.35, -0.1)$.

### 4.1.3.3 Physical $Q$-matrix eigenvalue distributions and RMT

To complete the spin $1/2$ analysis, we examine in more detail whether the RMT predictions derived in section 3 are verified in generic chaotic spin chain models. We already touched upon this question in Figure 3, where a qualitative agreement was found between the estimate (44) and the mean of physical $Q$-eigenvalue distributions. A quite similar qualitative behavior was in fact also observed for integrable models.

There is, however, a second characteristic property of chaotic models that integrable systems do not generally display at intermediate $k$: The sharp concentration about the mean value. This property was important in deriving a direct correlation between the mean of the $Q$-eigenvalue distribution and the saturation height of the complexity of chaotic models. In Figure 10, we show the shape of the $Q$-matrix distribution for considerably larger $N_{loc}$ than previously displayed. This compels the peak to depart from the right edge of the distribution. The RMT prediction for the variance (49) implies that, at fixed $N_{loc}/D^2$, the distribution concentrates about the mean (44). Since it is not always possible to keep $N_{loc}/D^2$ fixed in terms of varying locality specifications as $L$ increases, we proceed as follows. We fix the ratio $N_{loc}/D^2$ by considering the set of all generators, ordered in a systematic way by increasing locality, and putting a cutoff on $N_{loc}$ so as to select the first 15% elements in this set to be local. It is interesting to note that in this case the peak of the distributions in Figure 10 is very well predicted by (44). Therefore, while the estimate for the mean of the distribution (44) is reliable up to a factor of 2 at small $k$, as seen in Figure 3, the relative error in the estimate is much smaller for larger $N_{loc}/D^2$. In addition, the variance on the distribution clearly decreases as the dimension of the Hilbert space increases, as expected from (49). A close inspection of the precise values of the variance indicates that the agreement works at the level of the order of magnitude. In contrast, a similar exercise for the integrable Ising model leads to very different histograms, displayed on the right of Figure 10. The distributions are flatter than for the chaotic examples and a small peak at low values develops near the kernel.

Finally, we verify the relation (53) between the $Q$-matrix eigenvalues and the saturation height of chaotic models by comparing it with the observations for the Ising model. Figure 11 contains a comparison between the numerically computed saturation heights and the estimate (53) using the numerically computed mean of the $Q$-eigenvalue distribution, for the integrable

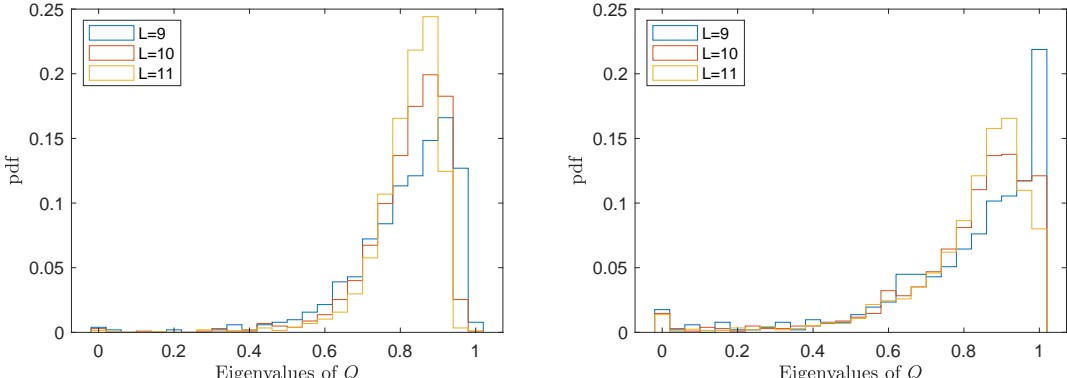

Figure 10: Histograms of the eigenvalues of the Q-matrix for **Left:** the chaotic Ising model (57) with $(h_x, h_z) = (-1.05, 0.5)$ for $L = 9, 10, 11$ and fixed $N_{loc}/D^2 = 0.15$. For the purpose of comparing the concentration properties of the distribution at varying $L$, we normalize the distributions so that the area below all the staircase curves is unity. The peak of the distribution can be seen to be close to 0.85, as expected from (44). **Right:** The same histograms for the transverse Ising at $(h_x, h_z) = (-1.05, 0)$, for comparison.

and chaotic Ising models at different locality thresholds. The chaotic systems can be seen to follow the estimate very closely, in contrast to the integrable instances. In particular, the integrable points move away from the line $\mathcal{C}_{sat} = \mathcal{C}_{est}$ as $k$ increases. This observation suggests that there are further mechanisms underlying complexity reduction in integrable systems beyond the stronger shift of the mean of the $Q$-eigenvalue distribution towards zero as $k$ increases, compared to chaotic systems. This is not surprising as the relation (53) between the mean of the $Q$-eigenvalue distribution and the complexity plateau height assumes concentration of the distribution, which is not necessarily present for integrable systems. In particular, Figure 11 confirms the hypothesis that the complexity reduction is largely a consequence of the additional zero eigenvalues of the $Q$-matrix connected to the locality properties of the tower of conservation laws.

We note that the chaotic points also appear slightly below the predicted line. We believe this is due to the fact that complete concentration can never occur in physical systems, even in the thermodynamic limit $L \to \infty$, since the Hamiltonian will always define an exact zero $Q$-eigenvalue. As a consequence, there is always at least one eigenvalue that does not participate in this behavior. One way to verify this intuition is to let the identity be a local generator. In this case, we observed that the chaotic data points indeed move slightly more away from the line $\mathcal{C}_{sat} = \mathcal{C}_{est}$. A complementary approach to verify the origin of this small discrepancy is to replace the Hamiltonian eigenbasis with a random matrix. For random matrices, we showed that concentration occurs, and hence the estimate (53) should be quite accurate. In addition, random matrices have no structure whatsoever when expanded in a basis of local and nonlocal generators, in contrast to the eigenbasis of physical Hamiltonians which are necessarily constrained by the locality properties of the Hamiltonian. Therefore, we do not expect the resulting complexity plateau value to be systematically lower than the estimate (53) as is observed for physical eigenbasis. This picture can indeed be verified numerically.

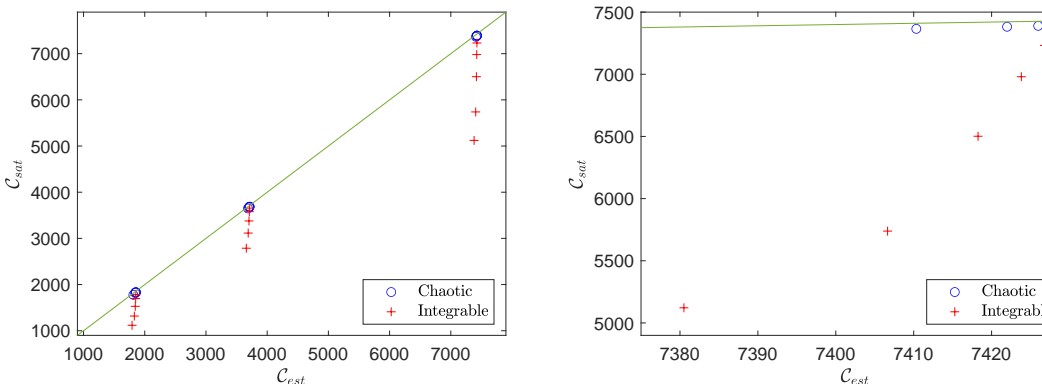

Figure 11: **Left:** The saturation height of the complexity bound against the estimate (53) for a few runs of the Ising model at $L = 10, 11, 12$ for $k_{sp} = k_{op} = 2, 3, 4, 5, 6$ for both the integrable case with $(h_x, h_z) = (-1.05, 0)$ and the chaotic case $(h_x, h_z) = (-1.05, 0.5)$. The green line is $\mathcal{C}_{sat} = \mathcal{C}_{est}$. **Right:** We zoom in on the data for $L = 12$.

## 4.2 Spin 1 chains

### 4.2.1 Locality parameters at $s = 1$

The locality specification of the generators (55) of spin systems at $s = 1/2$ relied on two parameters $k_{op}$ and $k_{sp}$. The freedom to increase the latter independently of the former allowed us to investigate how the addition of local directions that are not directly related to the integrability structure of the Hamiltonians (57) at $h_z = 0$ and (67) affects the complexity reduction. As we shall discuss next, increasing the spin representation at every site naturally introduces a third parameter in the characterization of the locality of generators. In this section, we analyze the relation between complexity reduction and integrability in this new setup.

Before discussing the Hamiltonians relevant to our analysis, we need to specify how to separate generators into easy and hard directions. Unlike at spin 1/2, linear combinations of the Pauli matrices do not generate the entire algebra of operators acting on a single site of the chain. Operators of the type (55) are therefore not sufficient to characterize the entire set of operators acting on the total spin chain Hilbert space. Nevertheless, there exists a completely systematic way of constructing a basis of Hermitian generators for an arbitrary spin quantum number $s$, at a single site, which can be heuristically motivated as follows.[18] A generic operator acting on a spin $s$ representation $V_s$ can be interpreted as a state in the tensor product of two spin $s$ representations $V_s \otimes V_s$

$$\sum_{m,m'} T_{mm'} |s, m\rangle \langle s, m'| \to \sum_{m,m'} T_{mm'} |s, m\rangle \otimes |s, m'\rangle, \tag{74}$$

with $-s \leq m, m' \leq s$ (where $m, m'$ increase in steps of 1). This tensor product splits into irreducible representations of spin $J = 0, 1, \cdots, 2s$. The spin 0 component of $T_{mm'}$ corresponds to the identity element on $V_s$, while the spin 1 representation in this decomposition is carried by the generalizations of the standard Pauli matrices:

$$(S_x)_{jk} = \delta_{j,k-1} \sqrt{j(2s - j + 1)} + \delta_{j-1,k} \sqrt{k(2s - k + 1)}, \tag{75}$$

$$(S_y)_{jk} = -i\delta_{j,k-1} \sqrt{j(2s - j + 1)} + i\delta_{j-1,k} \sqrt{k(2s - k + 1)}, \tag{76}$$

$$(S_z)_{jk} = \delta_{j,k} 2(s - j + 1), \tag{77}$$

---

[18]More details and the explicit construction of the basis elements can be found in [53].

where $j$ and $k$ run from 1 to $2s + 1$ and which satisfy the commutation relations (54). Each of the higher dimensional ($J > 1$) irreducible representations are in natural correspondence with $2J + 1$ linearly independent combinations constructed from products of $J$ generalized Pauli operators (75)-(77). A complete basis for the operators on $V_s$ based on this intuition, containing $(2s + 1)^2$ generators, can be straightforwardly found by means of specific linear combinations in terms of the Clebsch-Gordan coefficients which guarantee orthogonality of the resulting operators [53]. A MATLAB implementation of this systematic procedure can be found in [43].

One can subsequently obtain a complete basis of operators for an $L$-sized chain of spin $s$ degrees of freedom by taking tensor products of single site generators, as in the spin $1/2$ case (55). In addition to $k_{op}$ and $k_{sp}$, this description naturally introduces an 'internal' locality parameter, which characterizes the difficulty of implementing a single site operator. We henceforth define the internal locality degree $k_{int}$ of a spin chain generator as the largest number of generalized Pauli operators (75)-(77) (taking all sites into account) appearing in a single term of its expansion.

### 4.2.2 Integrable and non-integrable spin 1 Hamiltonians

Integrability would not be preserved in general with naive increase in the magnitude of spin, and the Hamiltonian needs to be cleverly adjusted under such increase. Since it is quite uncommon for a model, like the Heisenberg Hamiltonian, to be integrable for all choices of coupling constants, we restrict in the following to generalizing the model with the most symmetry to higher spin: The XXX chain. We first note that naively replacing the Pauli matrices appearing in the isotropic (67) by their spin 1 generalizations (75)-(77) breaks integrability, as anticipated. We shall therefore take the Hamiltonian

$$H_{\text{naive}} = \sum_j \left[ S_x^{(j)} S_x^{(j+1)} + S_y^{(j)} S_y^{(j+1)} + S_z^{(j)} S_z^{(j+1)} \right],$$ (78)

as a non-integrable representative.[19]

To generalize the Heisenberg spin chain to higher spin while maintaining integrability, it turns out beneficial to introduce an isotropic, $SU(M)$-symmetric Hamiltonian [54]

$$H_M = \sum_{j=1}^{L} t_a^{(j)} t_a^{(j+1)},$$ (79)

where the sum over repeated $a$ indices is implied. In (79), the matrices $t_a^{(j)}$ at site $j$, with $a = 1, ..., M^2 - 1$, form a set of generators for $SU(M)$, transforming in the adjoint representation and satisfying

$$[t_a^{(j)}, t_b^{(k)}] = 2i f_{ab}{}^c \, \delta_{jk} \, t_c^{(j)},$$ (80)

$$\{t_a^{(j)}, t_b^{(k)}\} = (4\delta_{ab}/M + 2d_{ab}{}^c t_c) \, \delta_{jk},$$ (81)

with $f_{ab}{}^c$ the structure constants of $SU(M)$ and $d_{ab}{}^c$ a completely symmetric tensor. For $M = 2$, this approach simply yields the spin $1/2$ XXX Hamiltonian. At $M = 3$, the matrices $t_a^{(j)}$ are the Gell-Mann matrices. These matrices are explicitly known in terms of spin 1 generalized Pauli

---

[19]We shall use the terminology 'non-integrable' as opposed to 'chaotic' when referring to (78) because we have not been able to conclude with certainty that the spectral statistics associated to (78) is Wignerian. A careful treatment of the nearest neighbor spacing statistics requires one to take the global conservation laws into account and the spectral analysis needs to be performed in each energy sector of fixed quantum numbers separately. Consequently, the sizes of the blocks originating from the large number of global symmetries of the Hamiltonian (78) at $L = 7$ are too small to provide meaningful statistics.

operators. Writing the Hamiltonian (79) in terms of (75)-(77), one finds that it includes the naive Hamiltonian (78) and introduces an additional term that renders the sum integrable:

$$H_{\text{int}} \equiv H_3 = \frac{1}{2} \sum_{j=1}^{L} \left[ x^{(j)} + \frac{1}{4} \left( x^{(j)} \right)^2 - \frac{16}{3} \right], \tag{82}$$

where

$$x^{(j)} = \sum_{a=x,y,z} S_a^{(j)} S_a^{(j+1)}. \tag{83}$$

It is interesting to remark that integrability appears to require $k_{int} = 4$, as opposed to (78) which has locality degree $k_{int} = 2$.

The Hamiltonian (82) should not be mistaken with the Fateev-Zamolodchikov spin chain [55], which is also integrable and resembles (82) except for one flipped sign. This modification breaks the $SU(3)$ symmetry down to $SU(2)$, and the two integrable Hamiltonians are inequivalent. A concise account of the different integrable systems that are constructed from (83) can found in the introduction of [56]. In the following, we shall exclusively use the $SU(3)$ symmetric Hamiltonian (82) since an explicit tower of conserved charges can be straightforwardly derived in this case [54].

Expressing the Hamiltonian (79) in terms of the generalized Pauli matrices makes an $SU(2)$ subgroup of the $SU(3)$ symmetry apparent. This reduced symmetry group is shared by the non-integrable Hamiltonian (78). Just as in the spin $1/2$ XXX Heisenberg model, the global charges, for both models, therefore include momentum conservation, symmetry under reflections, as well as (68) and (69) evaluated with the generalized spin matrices (75)-(77). The expression of the former two symmetry operators in terms of generalized Pauli matrices can be suitably extended to higher spins. Details of the implementation can be found in [43].

The global $SU(2)$ invariance group introduces many degeneracies in the spectrum. Before computing the enlarged $\tilde{Q}$-matrix, we shall therefore simultaneously diagonalize the non-integrable Hamiltonian (78) with the momentum operator $P$, $\mathcal{J}^z$ and $\mathcal{J}^2$ to speed up the calculation. In contrast, for the integrable Hamiltonian it is more natural to use the $SU(3)$ quantum numbers to split degeneracies. We therefore start by simultaneously diagonalizing the Hamiltonian with the momentum operator, the two $SU(3)$ analogues of $\mathcal{J}^z$ (built from the two Gell-Mann matrices that are diagonal in the standard basis) as well as the $SU(3)$ analog of $\mathcal{J}^2$. Note that, as with $\mathcal{J}^z$, two of these global charges are always local, with locality $k_{sp} = k_{op} = 1$ and $k_{int} = 2$, and will therefore appear as null directions of the $Q$-matrix for any locality threshold.

In addition to the global charges mentioned above, the integrable structure of (82) leads to a tower of conserved operators. These conservation laws are parameterized by $k$, which characterizes the term of largest locality degree

$$H_k = \sum_j \left( \left( \left( \mathbf{t}^{(j)} \times \mathbf{t}^{(j+1)} \right) \times \mathbf{t}^{(j+2)} \right) \times \cdots \times \mathbf{t}^{(j+k-2)} \right) \cdot \mathbf{t}^{(j+k-1)} + \dots, \tag{84}$$

in terms of the Gell-Mann matrices [51]. The cross-products are a shorthand notation for $(\mathbf{t}^{(j)} \times \mathbf{t}^{(k)})^a = f^{abc} t_b^{(j)} t_c^{(k)}$. For concreteness, we list the first two of these conserved charges:

$$H_3 = \sum_j f^{abc} t_a^{(j)} t_b^{(j+1)} t_c^{(j+2)}, \tag{85}$$

$$H_4 = \sum_j f^{abp} f^{pcd} t_a^{(j)} t_b^{(j+1)} t_c^{(j+2)} t_d^{(j+3)} + t_a^{(j)} t_a^{(j+2)}. \tag{86}$$

We note that, although not obvious, the locality degree of $H_3$ is $k_{op} = k_{sp} = 3$ and $k_{int} = 5$, while for $H_4$ we find $k_{op} = k_{sp} = 4$ and $k_{int} = 7$.

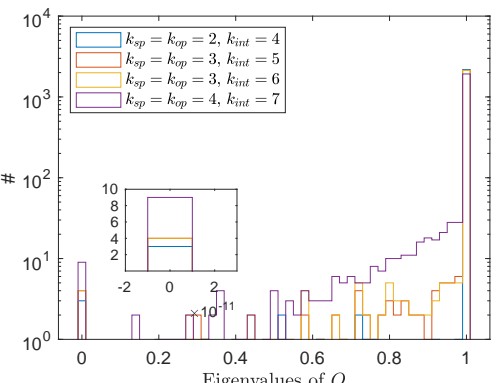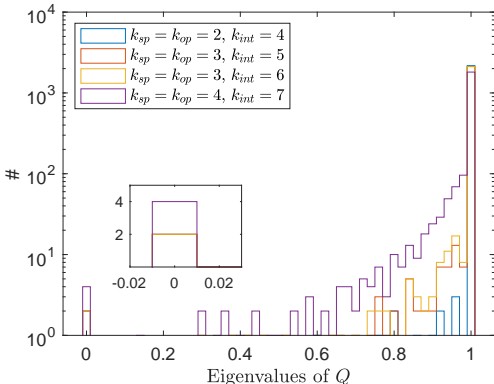

Figure 12: Histograms of the eigenvalues of the $Q$-matrix for **Left:** the integrable spin 1 XXX model (82) and **Right:** the non-integrable spin 1 XXX model (78) at $L = 7$.

### 4.2.3 Complexity reduction and integrability at $s = 1$

In Figure 12, we show the differences in $Q$-eigenvalue distributions for the integrable (82) and non-integrable (78) Hamiltonians. In this case, the non-integrable distribution is noticeably more spread out than what we found for the chaotic models at spin $1/2$. This may be due to the larger set of global symmetries. Indeed, as for the XXZ chain at $s = 1/2$, both Hamiltonians (78) and (82) commute with the conserved charge $\mathcal{J}^z$ whose locality is characterized by $k_{op} = k_{sp} = k_{int} = 1$. This global conserved charge therefore leads to an additional null direction of the $Q$-matrix in all the histograms displayed in Figure 12. Another possible explanation for the observed spread of the non-integrable distribution is that the model may not qualify as 'strongly chaotic'.

The clearest distinction between the right and left distributions therefore resides in the size of the $Q$-matrix kernel. The integrable model displays an increase in the number of null directions at $k_{int} = 5$ and $k_{int} = 7$, in accordance with the locality degree of the charges (85) and (86). The sudden increase in the size of the $Q$-matrix kernel at $k_{int} = 7$ in both dynamics can be understood as a consequence of the traceless version of the global charges $(\mathcal{J}^z)^2$ and $\mathcal{J}^2$ becoming local.

Finally, we compute the saturation values of the corresponding complexity curves and show the results in Figure 13. We find that the complexity of the integrable Hamiltonian dynamics displays a visible reduction already at the lowest possible locality threshold, which may be understood as originating from the additional local conservation law coming from the $SU(3)$ global symmetry group. In the right plot, we display the ratio of the integrable and the chaotic curve. From this, we conclude that the largest reductions in the ratio of the two curves occur when the number of exact null directions increases.

## 5 Conclusions

We have explored the upper bound on Nielsen's complexity of quantum evolution derived from the minimization problem (26), and revealed analytic connections between the reduction of this bound for integrable evolution, and towers of conservation laws inherent to integrable systems. This is the first time, to the best of our knowledge, to have such an explicit connection between analytic solvability and observable decrease in quantitative measures of quantum complexity.

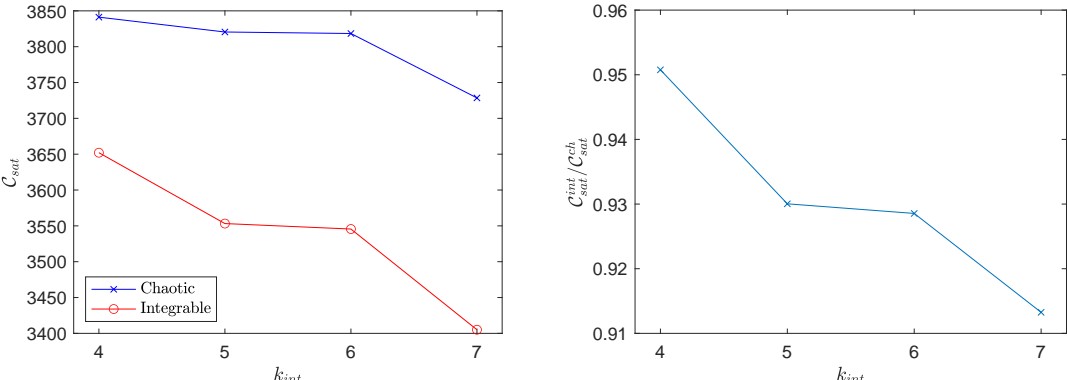

Figure 13: **Left:** Saturation values of the complexity bound as a function of $k_{int}$, with $k_{sp} = k_{op} = \lceil k_{int}/2 \rceil$, for the non-integrable (78) and integrable (82) spin 1 Hamiltonians. The length of the chain is set to $L = 7$. **Right:** In order to highlight the complexity reduction of the integrable model, we display the ratio between the integrable and non-integrable curves on the left plot.

Integrable systems are characterized by towers of conservation laws. It is very typical for these towers to be organized as analytic expressions whose sophistication grows at every next level of the tower (for example, they can be polynomials in the fundamental variables and their derivatives of higher and higher degree, or contain higher and higher derivatives, or involve terms structured as many-body interactions involving more and more particles). Such structures are visualized particularly naturally in those systems whose classical limits are Lax-integrable [57], since in that case, the conservation laws in the tower are expressed as traces of higher and higher powers of one of the Lax operators. They are, however, ubiquitously present in integrable systems, including those without straightforward classical limits, such as the spin chains we have treated here.

The notion of simpler (more local, few-body, etc) operators appears as well in the definition of Nielsen's complexity (11), which is where the first hint of our relation between integrability and complexity becomes apparent. In this definition, a majority of Hilbert space operators (defining 'hard' directions in the manifold of unitaries) are weighted by the penalty factor $\mu$, so that the curves whose lengths define Nielsen's complexity predominantly stay away from these directions. The physical qualities typically used to define the remaining 'easy' operators are very similar to what typically characterizes low-lying members of towers of conservation laws in integrable systems.

The final piece that completes the puzzle is the upper bound on Nielsen's complexity given by (26). This bound is in the form of minimizing a multivariate quadratic polynomial over an integer hypercubic lattice, with the polynomial being defined by the $Q$-matrix (27). This matrix has the property that its null eigenvalues exactly correspond to the conservation laws that belong to the class of 'easy' operators used to define Nielsen's complexity in (11). This property is visible in (30)-(31). At the same time, any null (small) eigenvalues of $Q$ create flat (nearly flat) directions in the lattice minimization problem (26), lowering the complexity estimate. Integrable systems, by definition, come with a large number of null vectors of $Q$, which lowers the complexity directly. Since eigenvalues of 'natural' large matrices tend to form continuous eigenvalue distribution, one may also legitimately expect that extra null eigenvalues will be accompanied by extra small eigenvalues, lowering the complexity further.

We have tested the above picture extensively using integrable spin chains (compared for contrast with generic, non-integrable spin chains). These systems are characterized by a rich variety of towers of conservation laws manifestly displaying the general properties outlined

above, see e.g. (63)-(66) and (84)-(86). Different classes of 'easy' operators could be defined according to their polynomial degree, the total number of lattice sites involved and spatial locality, and one could see that complexity reduction correlates with the number of conservation laws that fit into the given locality specifications in section 4.

In parallel with our main theme, which is the relation between integrability and complexity, we have reported two additional useful developments that cast further light on the complexity bound (26). First, in many analytically solvable models (but not in generic chaotic models), one expects degenerate energy levels. In such situations, the prescription for computing the complexity bound originally given in [6] is incomplete, since it needs to be supplemented with a choice of Hamiltonian eigenbasis that simultaneously diagonalizes other conserved quantities, *a priori* unknown. We designed, in section 2.2, a refinement of the approach in [6] that handles this issue and automatically produces the necessary basis. Second, we have explored the implications of random matrix theory for the $Q$-matrix in section 3 by assuming that the Hamiltonian eigenvectors form a random orthonormal basis. This has produced a faithful qualitative picture of the behavior of the $Q$-matrix (27) and the minimization problem (26) for generic (non-integrable) models, validated numerically in section 4.

Our considerations invite a number of further questions that we briefly summarize below: First, the height of complexity plateaus in terms of the upper bound (26) appears to be robustly captured in terms of the average distance from the lattice (present manifestly in the formula for the bound). It appears that computing the average distance from general lattices remains an open mathematical problem, with some beautiful partial results seen in [31, 58]. Improving such estimates would be very useful for understanding the complexity bound (26) and could result in analogs for integrable systems of our random matrix estimates for generic systems in section 3.

Second, it remains an open problem whether one could move closer to the actual Nielsen complexity from our upper bound estimate (26). Perhaps one could try paths on the manifold of unitaries made of a few pieces of the form $e^{iVt}$ rather than only one such piece. A stumbling block for this program is the need to optimize in very high-dimensional spaces, and it remains to be seen whether this optimization can be effectively implemented. Optimistically, one could hope that the estimates for complexity of integrable evolution will be further lowered by such refinements, while chaotic evolution will remain largely insensitive to them (simply reflecting in the plateau regime the average distance between two points on the group of unitaries).

Finally, it is very interesting how the complexity estimates could be performed for much higher values of the various locality thresholds. In this regime, one expects a proliferation of linearly independent local conservation laws (note that products of conservation laws at lower locality thresholds may enter the list of conservation laws at higher thresholds depending on the precise definitions). Unfortunately, to meaningfully implement such regimes, one would need to go to much higher values of the total number of particles, since it only makes sense to call $k$-body interactions local if $k$ is much smaller than the total number of particles/spins. But the number of Hilbert space dimensions is typically exponential in the number of particles/spins, quickly pushing one beyond the limit of computational resources. Thus, analytic insights would be necessary to address this problem. Again, optimistically, one can hope for specifications where the number of particles goes to infinity while the locality threshold continues to grow very slowly with the number of particles, such that the ratio between complexities of integrable and chaotic evolutions becomes considerably more dramatic in this regime.

Our results contribute to the expanding body of work aimed at characterizing chaotic and integrable dynamics. In the context of spin 1/2 chains, investigations of operator spreading using e.g. the OTOC have shown little (early-time) sensitivity to the properties of the dynamics [59–61]. In contrast, measures of the operator entanglement have been shown to successfully distinguish chaotic and integrable spin chains, see e.g. [62]. It would be interesting to

investigate the relation between operator entanglement growth and Nielsen's complexity in more detail, perhaps following the ideas of [63,64].

## Acknowledgments

**Funding information**

This research has been supported by FWO-Vlaanderen project G012222N and by Vrije Universiteit Brussel through the Strategic Research Program High-Energy Physics. OE has been supported by by Thailand NSRF via PMU-B (grant numbers B01F650006 and B05F650021). MDC is partially supported by the Simons Foundation Award number 620869 and by STFC consolidated grant ST/T000694/1. The resources and services used in this work were provided by the VSC (Flemish Supercomputer Center), funded by the Research Foundation - Flanders (FWO) and the Flemish Government.

## A  Weingarten functions

This appendix contains the unitary Weingarten functions relevant for the computations of the four- and eight-point functions of random unitary matrix elements drawn from the Gaussian Unitary Ensemble [42,65][20] performed in section 3:

$$\text{Wg}^U([1,1],D) = \frac{1}{D^2-1}, \tag{A.1}$$

$$\text{Wg}^U([2],D) = -\frac{1}{D(D^2-1)}, \tag{A.2}$$

$$\text{Wg}^U([1,1,1,1],D) = \frac{6-8D^2+D^4}{D^2(-36+49D^2-14D^4+D^6)}, \tag{A.3}$$

$$\text{Wg}^U([1,1,2],D) = -\frac{1}{9D-10D^3+D^5}, \tag{A.4}$$

$$\text{Wg}^U([2,2],D) = \frac{6+D^2}{D^2(-36+49D^2-14D^4+D^6)}, \tag{A.5}$$

$$\text{Wg}^U([1,3],D) = \frac{-3+2D^2}{D^2(-36+49D^2-14D^4+D^6)}, \tag{A.6}$$

$$\text{Wg}^U([4],D) = \frac{5}{D(-36+49D^2-14D^4+D^6)}, \tag{A.7}$$

where the square brackets in the first argument of the Weingarten functions denote the cycle type of the permutation $\tau\sigma^{-1}$ in (41). For instance, the permutation cycle $(1)(23)(4)$ has cycle type $[1,1,2]$.

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
