# Peer review of "Integrability and complexity in quantum spin chains"

_SciPost Physics, doi:SciPost Phys. 16, 041 (2024)_

## Round 1 · Referee Report · Anonymous (Referee 1) · 2023-9-29

Strengths

  1. Important and relevant results on the connection between complexity and dynamics.
  2. Clearly written. Also well contrasted with previous results in the literature (especially Ref. [6] by the same authors)
  3. Many practical examples for integrable and chaotic dynamics.

Weaknesses

No relevant ones.

Report

This manuscript proposes a new interesting technique to distinguish between chaotic and integrable models based on the concept of complexity. The topic of complexity in many body quantum systems is very timely and of fundamental importance. The same is also the case for the quest of setting apart chaotic and integrable dynamics by means of simple and computable quantities. It is very remarkable to establish a connection between these two subjects.

The manuscript is extremely well organized with a clear introduction where the main results are summarized. Sec. 2 contains the definitions of the quantities of interest. In Secs. 3 and 4 the main results for chaotic and integrable dynamics are derived. Many examples are reported to corroborate the results and ideas. I have no doubts concerning the publication of this interesting manuscript in Scipost Physcis. However, I would like that before the authors consider to comment on the point raised below.

In the literature, several other quantities to distinguish chaotic and integrable dynamics have been introduced and discussed. Among them, a very important and effective one is the local operator entanglement. Indeed it has been shown (see V. Alba, J. Dubail, and M. Medenjak, Phys. Rev. Lett. 122, 250603 (2019) and references therein) that such quantity grows at most logarithmically in time for integrable models while it grows linearly in chaotic ones. The authors should mention this fact and comment about possible connections between their results and the local operator entanglement.

Requested changes

See the report

---

## Round 1 · Referee Report · Anonymous (Referee 2) · 2023-11-13

Strengths

1 - tackles an ambitious and timely problem: complexity of chaotic vs. integrable dynamics

2 - focuses on a computable upper bound to complexity, which exhibits different behavior in chaotic/integrable models

3 - very detailed study of a very interesting object -the 'Q-matrix'- which allows to probe the local conservation laws and has implications on the complexity bound

4 - analytical results on the Q-matrix from random matrix

5 - extensive numerical checks in non-trivial interacting integrable spin chains

6 - very carefully written

Weaknesses

1 - complete lack of conciseness

2 - if one reads only this paper, then it is not quite clear what is new or not new here, compared to [SciPost Phys. 13, 090 (2022)] by the same authors

Report

Motivated by the notion of complexity of unitary evolutions, the authors study an upper bound on Nielsen's complexity -Eq. (2.24)- related to a matrix $Q$ of size $\dim(\mathcal{H}) \times \dim(\mathcal{H})$ defined for a subspace of 'easy' operators (such as local or few-body operators). The spectrum of the matrix $Q$ is claimed to be highly sensitive to integrability, in particular its kernel is directly tied to the existence of local conservation laws. The authors make general conjectures about properties of $Q$, which they relate to the behavior of their upper bound on complexity, especially to the plateau that the bound displays at long time.

The main claims are supported by analytical calculations for GUE random matrices, in the case of chaotic dynamics (Section 3). Extensive numerical checks are then provided both for chaotic and integrable spin chains (Section 4).

I find that the results are very substantial and exciting. They open a new pathway towards the fundamental goal of characterizing the impact of conservation laws on the complexity of quantum many-body dynamics. I also think they provide a synergetic link between research on complexity of quantum evolution operators, which so far has largely focused on chaotic models and/or fully connected models like the SYK model, and interacting integrable spin chains.

In my opinion, the only problem of this paper is the fact that it is too long. It took me a very long time to go through it, and while I feel that it was very instructive and inspiring, I am also a bit frustrated because the reading could have been faster and less painful if the manuscript were organized more clearly.

Several sections are extremely long, with no clear substructure. For instance, Section 2.1 is 10 pages long, presented as a whole block. It starts by discussing the definition of Nielsen's complexity, then some of its properties and equivalence/differences with other notions of complexity, then its geometric interpretation, then discusses a warm-up calculation of complexity in a simple case, then some generic properties of its short-time behavior, then its long-time behavior and plateau, in relation with typical distances to hybercube lattices, then refers to Ref. [6] for some observation about chaotic vs. integrable dynamics, then comes back to the general discussion of the problem with a penalty factor, then drops Eq. (2.24) -which plays a central role in the whole paper-, then discusses again distances in hypercubic lattices, then makes a long digression about some known numerical methods to solve the 'CVP', then repeats some information about distances in hypercubic lattices, then briefly mentions with Eq. (2.28) that the kernel of the matrix Q encodes the conservation laws of the model -a central point for the rest of the paper-, then discusses whether or not it is a good choice to declare that the identity operator is local, and then provides a loose discussion of integrability, repeating information that already appeared above, and finally briefly hints at the results of Sections 3 and 4. Please...

It would not be difficult to break this long section (and other similar sections, such as 4.1.3) that contain digressions and repetitions, into smaller, more focused, subsections or paragraphs, each with a clear title. In addition, it would help the reader to have a clear distinction between the results of Ref. [6] that are partially summarized in section 2.1, and the new results specific to this new paper.

In summary, I am happy to recomment publication of these results in Scipost Physics, but I would encourage the authors to try to make the manuscript a bit easier to read, if possible.

Requested changes

See above. If the authors could give more structure to the manuscript (subsections, paragraph titles, etc), and perhaps also trim it in order to avoid unnecessary repetitions, I think it could really make the manuscript more accessible.

Otherwise, the manuscript is very carefully written. I caught only a few typos:

  • before Eq. (3.10), it seems that a word is missing: perhaps 'We simplify the computation by working to leading order in $1/D$ and we define'

  • before Eq. (4.3): 'these non-trivial site' -> 'sites'

  • after Eq. (4.14): 'The former has an explicit U(1) symmetry' sounds weird, because both the former and the latter have that symmetry here

Finally, in the captions of the figures, it is said many times that what is shown is 'the complexity' (see e.g. caption of Fig. 6). It would be clearer to recall that it is the 'complexity bound' that is shown

---

## Round 2 · Referee Report · Anonymous (Referee 1) · 2023-12-14

Report

The authors carefully implemented the minor changes required by me and the other referee, so the manuscript is ready for publication in Scipost Physics.

---

## Round 2 · Referee Report · Anonymous (Referee 2) · 2024-1-3

Strengths

Same as in first report. With this new version, the readability has improved.

Report

The authors have provided satisfactory replies to my comments, and the presentation has improved, especially in Section 2.1. I am happy to recommend this new version for publication in Scipost Physics.

---

## Round 2 · List of Changes

We would like to thank both referees for their suggestions and constructive feedback. The manuscript has been updated with the following changes:

  • Section 2.1, the introduction to section 3 and section 4.2 have been restructured as requested.

  • We corrected the typos pointed out by the second referee, and changed the wording in a few places for clarity (including in the captions as requested).

  • We added a reference to [16] in footnote 1, as well as a new paragraph containing [59-64] at the end of the conclusions section.

In relation to the request by the second referee to clearly contrast what is new compared to [6], we refer to the subsection "statement of results" in the introduction, which highlights the new results of the present work.

Sincerely, B. Craps, M. De Clerck, O. Evnin, P. Hacker

---

## Editorial Decision

published